# Zero-Shot Trajectory Planning for Signal Temporal Logic Tasks

**Ruijia Liu**
Department of Automation
Shanghai Jiao Tong University
liuruijia@sjtu.edu.cn

**Ancheng Hou**
Department of Automation
Shanghai Jiao Tong University
hou.ancheng@sjtu.edu.cn

**Xiao Yu**
Institute of Artificial Intelligence
Xiamen University
xiaoyu@xmu.edu.cn

**Xiang Yin**
Department of Automation
Shanghai Jiao Tong University
yinxiang@sjtu.edu.cn

## Abstract

Signal Temporal Logic (STL) is a powerful specification language for describing complex temporal behaviors of continuous signals, making it well-suited for high-level robotic task descriptions. However, generating executable plans for STL tasks is challenging, as it requires consideration of the coupling between the task specification and the system dynamics. Existing approaches either follow a model-based setting that explicitly requires knowledge of the system dynamics or adopt a task-oriented data-driven approach to learn plans for specific tasks. In this work, we address the problem of generating executable STL plans for systems with unknown dynamics. We propose a hierarchical planning framework that enables zero-shot generalization to new STL tasks by leveraging only task-agnostic trajectory data during offline training. The framework consists of three key components: (i) decomposing the STL specification into several progresses and time constraints, (ii) searching for timed waypoints that satisfy all progresses under time constraints, and (iii) generating trajectory segments using a pre-trained diffusion model and stitching them into complete trajectories. We formally prove that our method guarantees STL satisfaction, and simulation results demonstrate its effectiveness in generating dynamically feasible trajectories across diverse long-horizon STL tasks. Project Page: https://cps-sjtu.github.io/Zero-Shot-STL/

## 1   Introduction

Signal Temporal Logic (STL) is a formal specification language used to describe the temporal behavior of continuous signals. It has become widely adopted for specifying high-level robotic behaviors due to its expressiveness and the availability of both Boolean and quantitative evaluation measures. Controlling robots under STL task constraints, however, is a challenging problem, as it requires balancing both the satisfaction of the task and the feasibility of the system dynamics. In cases where the environment and system dynamics are fully known, several representative methods have been developed, including optimization-based approaches [1, 2, 3], gradient-based techniques [4, 5], and sampling-based methods [6]. However, these methods are often difficult to apply in practical scenarios, where the system dynamics and environment are either unknown or difficult to model.

To address the challenge of unknown dynamics, several learning-based approaches have been proposed. One typical method is reinforcement learning (RL) [7, 8, 9, 10, 11, 12], where an appropriate reward function is designed to approximate the satisfaction of the STL task. However, these methods

39th Conference on Neural Information Processing Systems (NeurIPS 2025).

often struggle with long-horizon STL tasks and lack generalization capabilities across different tasks. Another approach involves first learning a system model and then integrating it with model-based planning methods. For example, in [13], the authors trained a neural network to approximate the system dynamics and combined it with an optimization-based approach. However, this method is limited to simple short-horizon STL tasks due to its high computational cost. In [14], the authors used goal-conditioned RL to train multiple goal-conditioned policies, referred to as "skills," to accomplish specific objectives. They then applied a search algorithm to determine the optimal sequence of "skills" needed to satisfy the given STL tasks. While this approach enables a certain degree of task generalizations, these tasks must be based on pre-defined objectives associated with the skills.

More recently, generative models, such as diffusion models [15], have emerged as a new approach for generating trajectories for systems with unknown dynamics [16, 17, 18, 19, 20], gaining popularity across various applications. Compared to traditional model-based reinforcement learning methods, these generative approaches are better suited for long-horizon decision-making and offer greater test-time flexibility [16], making them particularly effective for complex tasks. For example, for finite Linear Temporal Logic (LTL$_f$) tasks, [21] introduced a classifier-based guidance approach to steer the sampling of diffusion models, ensuring that generated trajectories satisfy LTL$_f$ requirements. Similarly, [22] proposed a hierarchical framework that decomposes co-safe LTL tasks into sub-tasks using hierarchical reinforcement learning. This framework employs a diffusion model with a determinant-based sampling strategy to generate diverse low-level trajectories, improving both planning success rates and task generalization.

In the context of STL trajectory planning, the use of generative models has also been explored recently. For example, [23] proposed a classifier-based guidance approach that leverages robustness gradients to guide diffusion model sampling, enabling the generation of vehicle trajectories that adhere to traffic rules specified by STL. Building on this, [24] introduced a data augmentation method to enhance trajectory diversity and improve rule satisfaction rates. However, these approaches are still limited to simpler STL tasks, primarily due to the complexity of optimizing robustness values and the inherent trade-off between maximizing reward objectives and maintaining the feasibility of the generated trajectories [25].

In this paper, considering that trajectories satisfying complex STL specifications are typically difficult to collect in real-world scenarios, we focus on composing such trajectories by stitching together short trajectory chunks. The main challenge lies in determining appropriate ways to combine these chunks such that the resulting trajectory satisfies the global STL specification while maintaining dynamic consistency and feasibility. Inspired by recent advances in decomposition-based STL planning[26], we propose a novel hierarchical framework that integrates task decomposition, search algorithms, and generative models. First, complex STL tasks are decomposed into a set of time-aware reach-avoid progresses. Next, a search algorithm, heuristically guided by the trajectory data, is employed to allocate these progresses and generate a sequence of waypoints with corresponding timestamps. Finally, a pre-trained diffusion model, trained on task-agnostic data, is used to sequentially generate trajectory chunks that connect adjacent timed waypoints. All trajectory chunks are then stitched together to form the complete trajectory.

To the best of our knowledge, our algorithm is the first data-driven approach with zero-shot generalization capability for complex STL tasks. We have formally proven the soundness of our STL decomposition and planning algorithm, which guarantees that the generated trajectories satisfy any given STL specification. Furthermore, we empirically evaluate the dynamic consistency and feasibility of the planned trajectories through simulation experiments. Simulation results demonstrate that our method achieves a high execution success rate across diverse long-horizon STL tasks, where the diffusion-based baseline fails. Moreover, it outperforms both the diffusion-based baseline and a standard non-data-driven method in planning efficiency.

## 2 Preliminaries

### 2.1 System Model

We consider a discrete time system with unknown dynamics

$$\mathbf{x}_{t+1} = f(\mathbf{x}_t, \mathbf{a}_t), \tag{1}$$

where $\mathbf{x}_t \in \mathbb{R}^n$ and $\mathbf{a}_t \in \mathbb{R}^m$ are the state and the action at time instant $t$, respectively. Given an initial state $\mathbf{x}_0$ and a sequence of actions $\mathbf{a}_0 \mathbf{a}_1 \ldots \mathbf{a}_{T-1}$, the resulting *trajectory* of the system is $\boldsymbol{\tau} = \mathbf{x}_0 \mathbf{a}_0 \mathbf{x}_1 \mathbf{a}_1 \ldots \mathbf{a}_{T-1} \mathbf{x}_T$, where $T$ is the horizon. The *signal* of the trajectory is referred to as the state sequence $\mathbf{s} = \mathbf{x}_0 \mathbf{x}_1 \ldots \mathbf{x}_T$ and we denote by $\mathbf{s}_t = \mathbf{x}_t \mathbf{x}_{t+1} \ldots \mathbf{x}_T$ the sub-signal starting from time step $t$.

## 2.2 Signal Temporal Logic

We use signal temporal logic (STL) to describe the formal task imposed on the generated state sequence [27]. Specifically, we consider STL formula in the Positive Normal Form (PNF) [28] whose syntax is as follows:

$$\varphi ::= \top \mid \mu \mid \varphi_1 \wedge \varphi_2 \mid \varphi_1 \vee \varphi_2 \mid \mathrm{F}_{[a,b]}\varphi \mid \mathrm{G}_{[a,b]}\varphi \mid \varphi_1 \mathrm{U}_{[a,b]}\varphi_2, \tag{2}$$

where $\top$ is the true predicate and $\mu$ is an atomic predicate associated with an evaluation function $h_\mu : \mathbb{R}^n \to \mathbb{R}$, i.e., predicate $\mu$ is true at state $\mathbf{x}_t$ iff $h_\mu(\mathbf{x}_t) \geq 0$. Furthermore, $\wedge$ and $\vee$ are logic operators "conjunction" and "disjunction", respectively; $\mathrm{U}_{[a,b]}, \mathrm{F}_{[a,b]}$ and $\mathrm{G}_{[a,b]}$ are temporal operators "until", "eventually" and "always", respectively; $[a, b]$ is a time interval such that $a, b \in \mathbb{Z}, 0 \leq a \leq b < \infty$. Note that, negation is not used in the PNF. However, as shown in [28], this does not result in any loss of generality as one can always redefine atomic predicates to account for the presence of negations, allowing any general STL formula to be expressed in PNF. In our work, we impose an additional restriction on the Prenex Normal Form of formulas. Specifically, for any formula of the form $\varphi_1 \mathrm{U}_{[a,b]}\varphi_2$, $\varphi_1$ can only involve temporal operator "always". This restriction is introduced for technical reasons, as it facilitates the decomposition of the overall formula into a set of progresses. For any signal $\mathbf{s} = \mathbf{x}_0 \mathbf{x}_1 \ldots \mathbf{x}_T$, we denote by $\mathbf{s}_t \vDash \varphi$ if $\mathbf{s}$ satisfies STL formula $\varphi$ at time $t$, and we denote by $\mathbf{s} \vDash \varphi$ if $\mathbf{s}_0 \vDash \varphi$. This is formally defined by the Boolean semantics of STL formulae as shown in **Appendix C.1**.

## 2.3 Planning with Unknown Dynamics

In the context of STL planning, the objective is to determine an action sequence such that the resulting signal satisfies the specified STL formula. When the system dynamics are perfectly known, this problem can be solved using model-based optimization approaches (e.g., see [1, 2, 3]). In contrast, our work addresses a setting with unknown dynamics. Specifically, we assume the mapping $f : \mathbb{R}^n \times \mathbb{R}^m \to \mathbb{R}^n$ is unknown, but a dataset of historical operational trajectories, consistent with the underlying unknown system dynamics, is available. Note that each trajectory in the dataset is collected from the previous task-agnostic operations and may vary in length. Our goal is to leverage these task-agnostic trajectories to generate new trajectories that satisfy any given STL formula, thereby achieving zero-shot task generalization at test time.

**Problem 1.** *Given a set of trajectories from the unknown system (1) and a STL formula $\varphi$ in the desired form, find a sequence of actions $\mathbf{a}_0 \mathbf{a}_1 \ldots \mathbf{a}_T$ such that the resulting signal $\mathbf{s}$ satisfies the STL formula, i.e., $\mathbf{s} \vDash \varphi$.*

# 3 Our Method

## 3.1 Overall Framework

First, we provide an overview of our proposed planning framework, whose overall structure is illustrated in Figure 1. Our method consists of three key components. The first is **semantics-based task decomposition**, in which the given STL formula is decomposed into a set of spatial-temporal *progresses* $\mathbb{P} = \mathbb{P}^{\mathcal{R}} \dot{\cup} \mathbb{P}^{\mathcal{I}}$, where $\mathbb{P}^{\mathcal{R}}$ denotes the set of *reachability progresses* and $\mathbb{P}^{\mathcal{I}}$ the set of *invariance progresses*. These progresses must be satisfied under a set of *time constraints* $\mathbb{T}$ defined over time variables $\Lambda$. The second component is **dynamics-aware progress allocation**. While the above decomposition is task-centric, ensuring that the progresses can be satisfied in the correct temporal order requires considering the system's underlying dynamics. To this end, we use a pre-trained *Time Predictor*, learned from task-agnostic trajectory data, to estimate the time required to transition between waypoints. A search-based algorithm then determines a sequence of timed waypoints $(\tilde{\mathbf{x}}_0, t_0), (\tilde{\mathbf{x}}_1, t_1), \ldots, (\tilde{\mathbf{x}}_n, t_n)$ such that each waypoint satisfies a corresponding reachability progress in $\mathbb{P}^{\mathcal{R}}$ and complies with all invariance progresses in $\mathbb{P}^{\mathcal{I}}$.

The third component is **trajectory generation**. Given the sequence of timed waypoints, we generate an executable trajectory that ensures each waypoint is visited at the correct time while satisfying all invariance progresses by stitching together shorter trajectory segments. We instantiate this using existing constrained sampling frameworks (e.g., SafeDiffuser [29]), which leverage diffusion models pre-trained on trajectory data to produce dynamically feasible, constraint-compliant trajectories through constrained sampling. Next, we provide the technical details of each component.

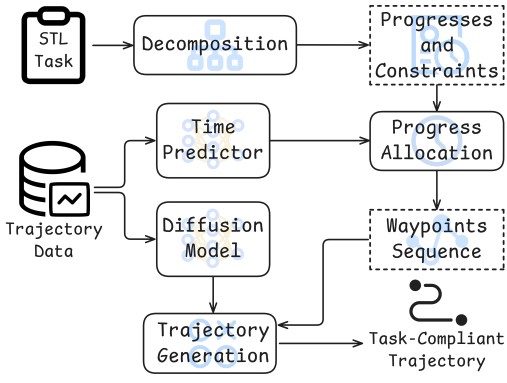

Figure 1: The Overall Framework of Our Proposed Method.

## 3.2 Decompositions of STL Formulae

**Eliminating Disjunctions.** For the given STL task $\varphi$, our first step is to covert it into the disjunctive normal form (DNF) $\tilde{\varphi} = \varphi_1 \vee \varphi_2 \vee \cdots \vee \varphi_n$, where each subformula $\varphi_i$ involves no "disjunction". Formally, for any STL formula, one can obtain its disjunctive normal form (DNF) by recursively applying the following replacements: (i) replace $\mathrm{F}_{[a,b]}(\varphi_1 \vee \varphi_2)$ with $\mathrm{F}_{[a,b]}\varphi_1 \vee \mathrm{F}_{[a,b]}\varphi_2$; (ii) replace $\mathrm{G}_{[a,b]}(\varphi_1 \vee \varphi_2)$ with $\mathrm{G}_{[a,b]}\varphi_1 \vee \mathrm{G}_{[a,b]}\varphi_2$; and (iii) replace $(\phi_1 \vee \phi_2)\mathrm{U}_{[a,b]}(\varphi_1 \vee \varphi_2)$ with $\bigvee_{i,j \in \{1,2\}} \phi_i \mathrm{U}_{[a,b]}\varphi_j$. Note that the last two replacements are not equivalent, making the resulting DNF $\tilde{\varphi}$ stronger than the original formula $\varphi$; i.e., $\tilde{\varphi} \Rightarrow \varphi$ but not conversely, adding conservativeness while preserving soundness. Furthermore, to achieve the task defined by $\tilde{\varphi}$, it suffices to satisfy one of the subformulas $\varphi_i$. Without loss of generality, we will assume henceforth that the DNF contains only a single subformula, as the STL planning problem can be addressed for each subformula individually. In other words, moving forward, we will focus on STL formulae, denoted directly by $\varphi$, without negations (due to the PNF) and without disjunctions (due to the DNF). Similar simplifications and assumptions are common in STL planning or control synthesis literature [30].

**Progresses.** We encode the STL task $\varphi$ by a finite set of *progresses* together with a set of *time-variable constraints*. For a signal fragment $\mathbf{s}_t = \mathbf{x}_t\mathbf{x}_{t+1}\ldots\mathbf{x}_T$, we use two canonical progress types:

- **Reachability progress** $\mathcal{R}(a_\Lambda + t, b_\Lambda + t, \mu)$: there exists $t' \in [a_\Lambda + t, b_\Lambda + t]$ such that $\mathbf{x}_{t'} \vDash \mu$.
- **Invariance progress** $\mathcal{I}(a_\Lambda + t, b_\Lambda + t, \mu)$: for all $t' \in [a_\Lambda + t, b_\Lambda + t]$ it holds that $\mathbf{x}_{t'} \vDash \mu$.

Unless stated otherwise we take $t = 0$ and write simply $\mathcal{R}(a_\Lambda, b_\Lambda, \mu)$ and $\mathcal{I}(a_\Lambda, b_\Lambda, \mu)$.

**Time Variables and Constraints.** The subscript "$\Lambda$" indicates that the interval endpoints may depend on a finite set of time variables $\Lambda_\varphi = \{\lambda_1, \ldots, \lambda_{|\Lambda|}\}$. A concrete assignment is a vector $\boldsymbol{\lambda} = [\lambda_1, \ldots, \lambda_{|\Lambda|}] \in \mathbb{Z}_+^{|\Lambda|}$. Throughout our decomposition (see Appendix D), each endpoint is an affine 0-1 combination of these variables (plus constants), hence we view $a_\Lambda, b_\Lambda : \mathbb{Z}_+^{|\Lambda|} \to \mathbb{Z}_+$. All time constraints are *unary* interval bounds of the form $\lambda_i \in [\underline{a}_i, \overline{b}_i]$; we collect them in a set $\mathbb{T}_\varphi$.

**Feasible Assignment Set.** The feasible set induced by $\mathbb{T}_\varphi$ is

$$\mathcal{F}_\varphi := \left\{ \boldsymbol{\lambda} \in \mathbb{Z}_+^{|\Lambda_\varphi|} \,\middle|\, \forall \mathfrak{t} \in \mathbb{T}_\varphi : \boldsymbol{\lambda} \vDash \mathfrak{t} \right\},$$

where $\boldsymbol{\lambda} \vDash (\lambda_i \in [\underline{a}_i, \overline{b}_i])$ iff the $i$-th component of $\boldsymbol{\lambda}$ lies in the specified interval. Because constraints are unary, $\mathcal{F}_\varphi$ factorizes coordinatewise.

**Satisfaction of Progresses under an Assignment.** Given $\boldsymbol{\lambda} \in \mathcal{F}_\varphi$, we abbreviate $\mathbf{s}_0 \vDash \mathcal{P}(\boldsymbol{\lambda})$ for the satisfaction of $\mathcal{P}(a_\Lambda(\boldsymbol{\lambda}), b_\Lambda(\boldsymbol{\lambda}), \mu)$, where $\mathcal{P} \in \{\mathcal{R}, \mathcal{I}\}$:

$$\mathbf{s}_0 \vDash \mathcal{R}(a_\Lambda(\boldsymbol{\lambda}), b_\Lambda(\boldsymbol{\lambda}), \mu) \iff \exists t \in [a_\Lambda(\boldsymbol{\lambda}), b_\Lambda(\boldsymbol{\lambda})] : \mathbf{x}_t \vDash \mu, \tag{3}$$

$$\mathbf{s}_0 \vDash \mathcal{I}(a_\Lambda(\boldsymbol{\lambda}), b_\Lambda(\boldsymbol{\lambda}), \mu) \iff \forall t \in [a_\Lambda(\boldsymbol{\lambda}), b_\Lambda(\boldsymbol{\lambda})] : \mathbf{x}_t \vDash \mu. \tag{4}$$

The full recursive decomposition procedure is detailed in **Appendix D**. To establish the soundness of the decomposition process with respect to the original STL specification, we propose the following lemma and the proof is provided in **Appendix G**.

**Lemma 1** (Soundness of Semantics-based STL Decomposition). *Let $\varphi$ be an STL formula in positive normal form (PNF) without disjunctions, and let $(\mathbb{P}_\varphi, \mathbb{T}_\varphi)$ along with the time-variable set $\Lambda_\varphi$ be the result of the recursive decomposition. Let $\mathcal{F}_\varphi$ denote the set of feasible assignments that satisfy all time constraints. Then, for any signal $\mathbf{s}_0$,*

$$\mathbf{s}_0 \vDash \varphi \iff \exists \boldsymbol{\lambda} \in \mathcal{F}_\varphi \ s.t. \ \forall \mathcal{P} \in \mathbb{P}_\varphi : \ \mathbf{s}_0 \vDash \mathcal{P}(\boldsymbol{\lambda}).$$

*In particular, if $\Lambda_\varphi = \varnothing$, i.e., no time variables appear in the decomposition procedure, the condition reduces to $\mathbf{s}_0 \vDash \varphi \iff \forall \mathcal{P} \in \mathbb{P}_\varphi : \ \mathbf{s}_0 \vDash \mathcal{P}$.*

### 3.3 Progress Allocation

**Basic Idea.** By Lemma 1, when system dynamics are ignored, STL planning reduces to a constraint-satisfaction problem over the decomposed progresses and time constraints $(\mathbb{P}_\varphi, \mathbb{T}_\varphi)$. Concretely, $\mathbf{s}_0 \vDash \varphi$ holds *iff* there exists a feasible time-variable assignment $\boldsymbol{\lambda} \in \mathcal{F}_\varphi$ such that every progress in $\mathbb{P}_\varphi$ is satisfied. This reformulation is advantageous for signal construction: instead of reasoning directly with temporal quantifiers in the original formula, it suffices to (i) ensure the *existence* of a feasible $\boldsymbol{\lambda}$, and (ii) construct a signal that satisfies all progresses under that assignment. We refer to this process as *progress allocation*.

However, unknown system dynamics pose additional challenges, as arbitrary allocation may result in dynamically infeasible plans. To address this, we adopt a search-based allocation algorithm, where the feasibility of each assignment is evaluated using a model (Time Predictor) learned from trajectory data, which implicitly captures the system dynamics.

**Preprocessing.** To perform the search-based allocation process, we further split the progresses $\mathbb{P}_\varphi$ as follows. For each invariance progress $\mathcal{I}(a_\Lambda, b_\Lambda, \mu) \in \mathbb{P}_\varphi$, we decompose it into a reachability progress $\mathcal{R}(a_\Lambda, a_\Lambda, \mu)$ and an invariance progress $\mathcal{I}(a_\Lambda + 1, b_\Lambda, \mu)$. For simplicity, we will denote the further decomposed progresses as $(\mathbb{P}, \mathbb{T})$ without subscripts, where $\mathbb{P} = \mathbb{P}^\mathcal{R} \cup \mathbb{P}^\mathcal{I}$. Due to this further decomposition, each invariance progress follows a unique reachability progress.

**Main Allocation Algorithm.** The main algorithm for progress allocation is presented in Algorithm 1. Note that the pseudo-code presents the search with an explicit stack for readability, whereas our reference implementation executes the same logic recursively.

The algorithm employs a depth-first search (DFS) to sequentially assign satisfaction times and waypoints for each reachability progress in $\mathbb{P}^\mathcal{R}$. When the algorithm terminates, it returns a sequence of waypoints with associated timestamps of form $(\tilde{\mathbf{x}}_0, t_0)(\tilde{\mathbf{x}}_1, t_1) \ldots (\tilde{\mathbf{x}}_n, t_n)$ such that each waypoint corresponds to the satisfaction of a reachability progress in $\mathbb{P}^\mathcal{R}$. During the search process, we maintain the current state $\mathbf{x}$, the current time step $t$, the set of remaining reachability progresses $\mathbb{P}^\mathcal{R}$, the set of all time constraints $\mathbb{T}$, and the searched sequence of waypoints with associated timestamps $\tilde{\mathbf{s}}$.

At each step, we select a remained reachability progress $\mathcal{R}(a_\Lambda, b_\Lambda, \mu)$ from $\mathbb{P}^\mathcal{R}$ as the next target progress from the current state $(\mathbf{x}, t)$. Specifically, to achieve progress $\mathcal{R}(a_\Lambda, b_\Lambda, \mu)$, function SampleState is used to determine a satisfaction time $t'$ and a corresponding new state $\mathbf{x}'$ such that $\mathbf{x}' \models \mu$. Then the searched waypoint $(\mathbf{x}', t')$ is appended to $\tilde{\mathbf{s}}$ and we remove progress $\mathcal{R}(a_\Lambda, b_\Lambda, \mu)$ from $\mathbb{P}^\mathcal{R}$. Furthermore, the time constraints $\mathbb{T}$ are updated based on the assigned state $\mathbf{x}'$ and time $t'$ according to function

---

**Algorithm 1** Main-Allocation

**Input:** Initial state $\mathbf{x}_0$, reachability progresses $\mathbb{P}^\mathcal{R}$, invariance progresses $\mathbb{P}^\mathcal{I}$, time variable constraints $\mathbb{T}$
**Output:** A valid waypoints sequence $\tilde{\mathbf{s}}$ or None if no solution is found
1: **Initialize:**
2: current state $\mathbf{x} \leftarrow \mathbf{x}_0$; current time $t \leftarrow 0$
3: sequence $\tilde{\mathbf{s}} \leftarrow [(\mathbf{x}, t)]$
4: *stack* $\leftarrow [(\mathbf{x}, t, \mathbb{P}^\mathcal{R}, \mathbb{T}, \tilde{\mathbf{s}})]$
5: **while** *stack* is not empty **do**
6:     $(\mathbf{x}, t, \mathbb{P}^\mathcal{R}, \mathbb{T}, \tilde{\mathbf{s}}) \leftarrow pop(stack)$
7:     **if** $\mathbb{P}^\mathcal{R} = \emptyset$ **then**
8:       **return** $\tilde{\mathbf{s}}, \mathbb{T}$     // All reachability progresses satisfied
9:     **for** each progress $\mathcal{R}(a_\Lambda, b_\Lambda, \mu) \in \mathbb{P}^\mathcal{R}$ **do**
10:       $t', \mathbf{x}' \leftarrow$ SampleState$(\mathcal{R}(a_\Lambda, b_\Lambda, \mu), \mathbf{x}, t, \mathbb{T}, \mathbb{P}^\mathcal{I})$
11:       **if** $t' \neq$ None **then**
12:         $\tilde{\mathbf{s}}' \leftarrow \tilde{\mathbf{s}}.(\mathbf{x}', t')$
13:         $\mathbb{P}^{\mathcal{R}'} \leftarrow \mathbb{P}^\mathcal{R} \setminus \{\mathcal{R}(a_\Lambda, b_\Lambda, \mu)\}$
14:         $\mathbb{T}' \leftarrow$ UpdateConstraint$(a_\Lambda, b_\Lambda, \mathbb{T}, \mathbf{x}', t')$
15:         // Push new state onto the stack
16:         *push* $(\mathbf{x}', t', \mathbb{P}^{\mathcal{R}'}, \mathbb{T}', \tilde{\mathbf{s}}')$ **onto** *stack*
17:     **end for**
18: **end while**
19: **return** None    // No valid sequence found

---

`UpdateConstraint`. Finally, the algorithm proceeds to the next iteration. If $\mathbb{P}^{\mathcal{R}}$ becomes empty, it indicates that all reachability progresses have been successfully assigned a satisfaction time. If no feasible assignment can be made, the algorithm backtracks to explore alternative assignments.

**Dynamic Maintenance of Time Variable Constraints.** During allocation, we dynamically maintain the set of time variable constraints $\mathbb{T}$. Once the satisfaction time of a reachability progress is determined, additional constraints are added accordingly. This enables the planner to query $\mathbb{T}$ at any point to determine potential ranges of $a_\Lambda$ and $b_\Lambda$ (we denote by $[a_{\Lambda,\mathbb{T}}^{\min}, a_{\Lambda,\mathbb{T}}^{\max}], [b_{\Lambda,\mathbb{T}}^{\min}, b_{\Lambda,\mathbb{T}}^{\max}]$) in a progress $\mathcal{P}(a_\Lambda, b_\Lambda, \mu)$ by solving integer linear programs (ILP), while ensuring consistency with previously assigned progresses. For example, to obtain $a_{\Lambda,\mathbb{T}}^{\min}$, we minimize $a_\Lambda(\boldsymbol{\lambda})$ subject to all constraints in $\mathbb{T}$.

Note that only the constraints are updated during planning; the time variable set $\Lambda$ remains fixed. As new constraints are introduced, the feasible assignment set $\mathcal{F}$ becomes increasingly restricted. If $\mathcal{F}$ becomes empty-i.e., no assignment $\boldsymbol{\lambda}$ satisfies all constraints-the current allocation is deemed infeasible, and backtracking is triggered to revise earlier decisions.

**Heuristic Order.** In the DFS of Algorithm 1, reachability progresses from $\mathbb{P}^{\mathcal{R}}$ are selected based on a heuristic order: progresses with earlier potential deadlines $b_{\Lambda,\mathbb{T}}^{\min}$ are prioritized; if multiple candidates share the same deadline, the one with the earlier potential start time $a_{\Lambda,\mathbb{T}}^{\min}$ is preferred.

**Constraint Update.** Let $\mathcal{R}(a_\Lambda, b_\Lambda, \mu)$ be the selected progress and $(\mathbf{x}', t')$ the assigned waypoint with time. We then add the following time constraints to $\mathbb{T}$:

$$a_\Lambda \leq t' \quad \text{and} \quad b_\Lambda \geq t'. \tag{5}$$

These constraints ensure that the progress can be satisfied at time $t'$. Recall that in our decomposition, each original invariance progress $\mathcal{I}(a_\Lambda, b_\Lambda, \mu)$ is split into a reachability progress $\mathcal{R}(a_\Lambda, a_\Lambda, \mu)$ and a residual invariance progress $\mathcal{I}(a_\Lambda + 1, b_\Lambda, \mu)$. Thus, if constraints (5) are added for a reachability progress of the form $\mathcal{R}(a_\Lambda, a_\Lambda, \mu)$, the value of $a_\Lambda$ is fixed to $t'$. Let $\mathbb{P}_{\text{det}}^{\mathcal{I}}$ denote the set of invariance progresses whose starting times have been determined. For any $\mathcal{I}(c, d_\Lambda, \mu) \in \mathbb{P}_{\text{det}}^{\mathcal{I}}$, if $\mathbf{x}' \not\models \mu$, we add an additional constraint $d_\Lambda < t'$ to ensure that the invariance progress terminates before $t'$, thereby avoiding conflicts with the assigned waypoint.

**Sample Timed Waypoints.** When a reachability progress $\mathcal{R}(a_\Lambda, b_\Lambda, \mu)$ is selected, we use function `SampleState` to determine a valid satisfaction time and waypoint state for this progress while ensuring compliance with invariance progresses.

The pseudocode of this function is shown in Algorithm 2. The process starts by computing the largest possible time interval $[t_{\min}, t_{\max}]$ for the reachability progress. Then the algorithm attempts to sample a candidate state $\mathbf{x}'$ with the satisfaction region of $\mu$ up to $N_{\max}$ times. For each attempt, we first calculate the minimum possible conflict time interval, denoted by $\mathcal{O}$, during which $\mathbf{x}'$ conflicts with the invariance progresses.

Once the conflict time interval $\mathcal{O}$ is computed, we further use function `TimePredict` to predict the arrival time $t'$ from the current state $\mathbf{x}$ to the sample state $\mathbf{x}'$. Particularly, if (i) $t' > t_{\max}$; or (ii) the feasible interval $[\max(t', t_{\min}), t_{\max}]$ is fully occupied by conflicting intervals in $\mathcal{O}$, then it means that the sample state $\mathbf{x}'$ is not feasible and we proceed to the next attempt. Otherwise, the earliest available time $t_{\text{new}}$ in the feasible interval is assigned as the satisfaction time, and the algorithm returns $t_{\text{new}}$ along with the sampled state $\mathbf{x}'$.

---

**Algorithm 2** `SampleState`

---

**Input:** reachability progress $\mathcal{R}(a, b, \mu)$, current state $\mathbf{x}$, time step $t$, time variable constraints $\mathbb{T}$, invariance progresses $\mathbb{P}^{\mathcal{I}}$

**Output:** Assigned satisfaction time $t_{\text{new}}$ of constraint $\mathcal{R}(a, b, \mu)$ and new state $\mathbf{x}'$ or `None` if no solution is found

1: $t_{\min} \leftarrow a_{\Lambda,\mathbb{T}}^{\min}, t_{\max} \leftarrow b_{\Lambda,\mathbb{T}}^{\max}$
2: **for** up to $N_{\max}$ attempts **do**
3:     Sample state $\mathbf{x}'$ such that $\mathbf{x}' \models \mu$
4:     **Initialize:** conflict time interval $\mathcal{O} \leftarrow \emptyset$
5:     **for all** $\mathcal{I}(c, d_\Lambda, \mu') \in \mathbb{P}_{det}^{\mathcal{I}}$ with determined starting time **do**
6:         **if** $\mathbf{x}' \not\models \mu'$ **then**
7:             $\mathcal{O} \leftarrow \mathcal{O} \cup [c, d_{\Lambda,\mathbb{T}}^{\min}]$
8:         **end if**
9:     **end for**
10:     $t' \leftarrow t + $ `TimePredict`$(\mathbf{x}, \mathbf{x}')$
11:     **if** $t' > t_{\max}$ **or** $[\max\{t', t_{\min}\}, t_{\max}] \setminus \mathcal{O} = \emptyset$ **then**
12:         **Continue** to next sampling attempt
13:     **end if**
14:     $t_{\text{new}} \leftarrow$ earliest time in $[\max\{t', t_{\min}\}, t_{\max}] \setminus \mathcal{O}$
15:     **return** $t_{\text{new}}, \mathbf{x}'$
16: **end for**
17: **return** `None`    // No valid time found

---

**Remark 1.** *When calculating the conflict time interval $\mathcal{O}$, we only consider invariance progresses whose starting times are determined. This is because, in the preprocessing process, we make each invariance progress follows a "preceding" reachability progress. If the starting time of an invariance progress is not determined, then it implies that its "preceding" reachability progress has not yet been satisfied and will only be satisfied strictly later than the current reachability progress. Consequently, this invariance progress will also start strictly later.*

**Prediction of Reachability Time.** In Algorithm 2, model `TimePredict` is used to estimates the time step (trajectory length) needed to transition from the current state $\mathbf{x}$ to the new state $\mathbf{x}'$. This model is trained on the same trajectory data, that will be used to train the Diffusion model. It assumes that the trajectory length between two states follows a Gaussian distribution. A simple multilayer perceptron (MLP) is used as the backbone of `TimePredict` model, and it is trained to predict the mean and variance of the trajectory length using a negative log-likelihood loss function. To account for tasks with avoidance requirements, which may require longer trajectories, a scaling factor $\gamma$ is applied to the predicted mean trajectory length. This factor allows control over the algorithm's conservativeness by adjusting the predicted trajectory length as needed.

**Soundness of the Progress Allocation Algorithm.** We establish the soundness of the progress allocation algorithm in Lemma 2, with proof provided in **Appendix G**.

**Lemma 2** (Soundness of Progress Allocation Algorithm). *Let $(\mathbf{x}_0, \mathbb{P}^{\mathcal{R}}, \mathbb{P}^{\mathcal{I}}, \mathbb{T})$ be the input to Algorithm 1. Assume the depth-first allocation terminates with a waypoint sequence $\tilde{\mathbf{s}} = (\tilde{\mathbf{x}}_0, t_0)\,(\tilde{\mathbf{x}}_1, t_1)\,\ldots\,(\tilde{\mathbf{x}}_n, t_n), \quad 0 = t_0 \leq t_1 \leq \cdots \leq t_n$, and let $\mathbb{T}_f$ be the final set of time-variable constraints with feasible assignment space $\mathcal{F}_f$.*

*Then $\mathcal{F}_f \neq \varnothing$, and for any signal $\mathbf{s}_0 = \mathbf{x}_0\mathbf{x}_1\ldots\mathbf{x}_T$ with $T \geq t_n$ and $\mathbf{x}_{t_i} = \tilde{\mathbf{x}}_i$ for $0 \leq i \leq n$, there exists $\boldsymbol{\lambda} \in \mathcal{F}_f$ such that*

$$\begin{cases} \forall \mathcal{R}(a_\Lambda, b_\Lambda, \mu) \in \mathbb{P}^{\mathcal{R}} : \ \mathbf{s}_0 \models \mathcal{R}(a_\Lambda(\boldsymbol{\lambda}), b_\Lambda(\boldsymbol{\lambda}), \mu), \\ \forall \mathcal{I}(a_\Lambda, b_\Lambda, \mu) \in \mathbb{P}^{\mathcal{I}}, \ \forall t \in [a_\Lambda(\boldsymbol{\lambda}), b_\Lambda(\boldsymbol{\lambda})] \cap \{t_i\}_{i=0}^n : \ \mathbf{x}_t \models \mu. \end{cases}$$

In other words, by interpreting the returned waypoints as key states within the signal $\mathbf{s}_0$, the algorithm guarantees that, under some feasible assignment of time variables, all reachability progresses are satisfied and no invariance progress is violated at any selected waypoint.

## 3.4 Trajectory Generation

Lemma 2 ensures that the signal constructed from the waypoints returned by the Progress Allocation algorithm satisfies all reachability progresses and does not violate any invariance progress at the selected time steps. We now address the problem of completing the full system trajectory so that the entire signal satisfies all progress conditions. Suppose the Progress Allocation algorithm returns a waypoint sequence $\tilde{\mathbf{s}} = (\tilde{\mathbf{x}}_0, t_0)\,(\tilde{\mathbf{x}}_1, t_1)\,\ldots\,(\tilde{\mathbf{x}}_n, t_n), \quad 0 = t_0 \leq t_1 \leq \cdots \leq t_n$. Our goal is to generate a system trajectory $\boldsymbol{\tau} = \mathbf{x}_0\mathbf{a}_0\mathbf{x}_1\mathbf{a}_1\ldots\mathbf{a}_{T-1}\mathbf{x}_T$ under system dynamics (1), whose corresponding signal $\mathbf{s}_0 = \mathbf{x}_0\mathbf{x}_1\ldots\mathbf{x}_T$ satisfies:

$$\begin{cases} T \geq t_n, \\ \mathbf{x}_{t_i} = \tilde{\mathbf{x}}_i \quad \text{for all } i \in \{0, \ldots, n\}, \\ \forall \mathcal{I} \in \mathbb{P}^{\mathcal{I}}, \ \mathbf{s}_0 \models \mathcal{I}. \end{cases} \tag{6}$$

Note that the time intervals of all invariance progresses are now fixed by selecting a specific feasible assignment $\boldsymbol{\lambda}^{\star} \in \mathcal{F}_f$. A special case arises when certain invariance progresses have not yet terminated at time $t_n$. In this situation, we simply wait at $\tilde{\mathbf{x}}_n$ until all invariance progresses end, since according to Lemma 2, $\tilde{\mathbf{x}}_n$ satisfies all predicates corresponding to the active invariance progresses at that time.

To solve this conditional trajectory generation problem, we employ a diffusion model trained on task-agnostic trajectory data, which enables the generation of dynamically feasible trajectories. Moreover, by incorporating constraints into the sampling process, the generated trajectories can be aligned with additional task requirements. However, directly generating long trajectories is challenging, as they often deviate from the training distribution. To mitigate this, we adopt a trajectory stitching strategy, generating individual segments $\boldsymbol{\tau}_i$ between each pair of adjacent waypoints $(\mathbf{x}_i, t_i)$ and $(\mathbf{x}_{i+1}, t_{i+1})$, and concatenating them to form the final trajectory $\boldsymbol{\tau}$. Specially, each segment must

satisfy the following conditions: (i) the length is $t_{i+1} - t_i + 1$, (ii) it starts at state $\mathbf{x}_i$ and ends at state $\mathbf{x}_{i+1}$, and (iii) it satisfies all invariance progresses active within the time interval $[t_i, t_{i+1}]$.

The diffusion model implementation is detailed in **Appendix E**. Briefly, the trajectory length is controlled via the length of initial noise, leveraging architectural properties of the backbone network [16]. Endpoint conditions are handled as an inpainting problem [16], by fixing the start and end states at every denoising step. To enforce invariance progresses, each $\mathcal{I}(a, b, \mu_i)$ is translated into a set of pointwise state constraints $h_{\mu_i}(\mathbf{x}_t) \geq 0$ for $t = a, \ldots, b$, commonly referred to as *safety constraints*. These constraints can be handled using various existing constrained sampling methods for diffusion models [31, 29, 32, 33, 34, 35]. We employ `SafeDiffuser` [29], which integrates control barrier functions (CBFs) [36] to enforce these constraints within the sampling process, providing formal guarantees under mild conditions.

**Soundness Analysis.** We establish the soundness of the overall framework in the following theorem. The proof is provided in **Appendix G**.

**Theorem 1** (Soundness of the Overall Planner). *Let $\varphi$ be an STL formula satisfying all assumptions in Sections 2.2 and 3.2. Assume that the* Semantics-based Decomposition *terminates with a finite set of progresses and constraints, the* Progress Allocation *algorithm returns a sequence of timed waypoints and a non-empty feasible assignment set, and the* Trajectory Generation *module produces a trajectory $\boldsymbol{\tau}$ whose signal $\mathbf{s}_0$ visits all timed waypoints and satisfies all active invariance progresses. Then the resulting signal satisfies the STL specification: $\mathbf{s}_0 \models \varphi$.*

## 3.5 Action Sequence and Control Protocol

Following mainstream diffusion-planner practice, we use the diffusion model solely to generate the state sequence; since action sequences are high-frequency and less smooth [17], we recover them via an learned inverse dynamics model [37] or with controllers that track the generated state sequence during execution. To guarantee strict adherence to the temporal windows imposed by STL tasks, we adopt a *time-synchronous control protocol*: each planning step is aligned with a fixed number $k$ of control steps, where $k$ is specified at task definition time to relate the planning horizon to the control frequency. At each planning step, the controller or inverse dynamics model therefore outputs exactly $k$ actions and executes $k$ control updates, ensuring that the temporal structure of the trajectory is preserved at runtime; unless otherwise stated, we set $k = 1$, so each planning step corresponds to precisely one control step.

## 4 Case Study

To illustrate the workflow of our algorithm, we consider a sequential visit and region avoidance task in the Maze2D (Large) environment [38]. In this scenario, the agent starts at the yellow point shown in Figure 2 and aims to complete the following STL task: $F_{[0,35]}(\mu_1 \wedge (F_{[35,45]}(\mu_2 \wedge F_{[10,30]}\mu_3))) \wedge G_{[0,110]}(\neg\mu_4 \wedge \neg\mu_5)$, where the predicate $\mu_i$ represents "reach region $\mu_i$," and $\neg\mu_i$ denotes "avoid region $\mu_i$". Intuitively, this STL task requires the agent

Table 1: STL Task Decomposition Results.

| Type | Details |
|---|---|
| $\mathbb{P}^{\mathcal{R}}$ | $\mathcal{R}(\lambda_1, \lambda_1, \mu_1)$, $\mathcal{R}(\lambda_1 + \lambda_2, \lambda_1 + \lambda_2, \mu_2)$, $\mathcal{R}(\lambda_1 + \lambda_2 + \lambda_3, \lambda_1 + \lambda_2 + \lambda_3, \mu_3)$, $\mathcal{R}(0, 0, \neg\mu_4)$, $\mathcal{R}(0, 0, \neg\mu_5)$ |
| $\mathbb{P}^{\mathcal{I}}$ | $\mathcal{I}(1, 110, \neg\mu_4)$, $\mathcal{I}(1, 110, \neg\mu_5)$ |
| $\mathbb{T}$ | $\lambda_1 \in [0, 35]$, $\lambda_2 \in [35, 45]$, $\lambda_3 \in [10, 30]$ |

to sequentially visit circular regions $\mu_1$, $\mu_2$, and $\mu_3$ within specific time intervals while avoiding regions $\mu_4$ and $\mu_5$ throughout the entire episode.

The environment layout and target regions are depicted in Figure 2. Our method leverages only an STL task-agnostic trajectory dataset to train the diffusion model, without prior knowledge of the map or system dynamics. The decomposed progresses and time constraints are summarized in Table 1. The planning algorithm then assigns completion times to each reachability progress, as indicated by the numbers next to start point and regions $\mu_1$, $\mu_2$, and $\mu_3$ in Figure 2, resulting in a sequence of waypoints with corresponding times. The diffusion model is then used to sequentially generate trajectories between adjacent waypoints while ensuring all invariance progresses are satisfied. The final generated trajectory is shown in the left subfigure of Figure 2. To evaluate the feasibility of the planned trajectory, we employ a simple PD controller to track it as described in Section 3.5.

The resulting execution trajectory is shown in the right subfigure of Figure 2. Using the open-source library `stlpy` [2], we calculate the robustness values for both the planned and actual execution trajectories as 0.180 and 0.115, respectively. Since both values are positive, the trajectories satisfy the STL task requirements.

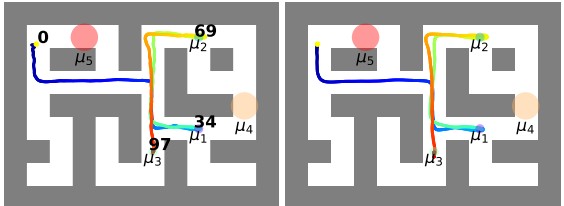

Figure 2: Planned Trajectory (left) and Actual Execution Trajectory (right) in Case Study.

## 5    Experiments

Although Theorem 1 guarantees the STL satisfaction of the trajectory, the use of a diffusion model to generate trajectories for systems with unknown dynamics means we cannot guarantee dynamic feasibility. Furthermore, our algorithm is not complete and exhibits some conservatism. Therefore, in our experiments, we primarily focus on evaluating the dynamic feasibility of the trajectories and the conservatism of the planning algorithm. To this end, we design several experiments to test these two aspects. All experiments were run on a PC with an Intel i7-13700K CPU and Nvidia 4090 GPU.

### 5.1    Experiment in Maze2D Environment

**Experimental Settings.**    We evaluate our algorithm in three Maze2D environments (U-Maze, Medium, and Large) to assess its zero-shot generalization capability on randomly generated STL tasks. The agent, starting from a random position, must complete the STL task by reaching the target regions within specified time intervals. To generate random STL tasks, we design nine templates (Table F.1) and randomly generate time intervals and atomic predicate regions for each template, resulting in 150 randomized tasks per environment. We compare our method with the Robustness Guided Diffuser (RGD) [23], which uses a diffusion model with robustness value gradient guidance to optimize the robustness value of trajectory. Both algorithms use diffusion models trained on the D4RL dataset [38] to generate trajectories and a PD controller for trajectory execution.

Table 2: Partial Result of Experiment in Maze2D. U:U-Maze; M:Medium; L:Large.

| Env | Type | Success Rate(%)↑ | | Robustness Value↑ | | Total Planning Time(s)↓ | | T1(s) |
|-----|------|------|------|------|------|------|------|------|
| | | RGD | ours | RGD | ours | RGD | ours | |
| U | 1 | 80.00 | 97.33 | 0.1084±0.1132 | 0.1938±0.0715 | 13.43±1.51 | 0.86±0.13 | 0.86 |
| | 2 | 36.67 | 92.00 | -0.2208±0.2826 | 0.1504±0.0814 | 16.65±2.06 | 0.64±0.17 | 0.64 |
| | 3 | 32.00 | 91.33 | -0.1695±0.2521 | 0.1354±0.0745 | 19.68±11.76 | 1.31±0.15 | 1.21 |
| M | 1 | 70.00 | 94.67 | 0.0885±0.2768 | 0.3205±0.0957 | 53.90±5.78 | 3.74±0.34 | 3.74 |
| | 2 | 34.67 | 89.33 | -0.2277±0.3089 | 0.2013±0.1950 | 70.69±8.27 | 2.72±0.67 | 2.72 |
| | 3 | 35.33 | 83.33 | -0.2393±0.3130 | 0.1761±0.2060 | 129.87±27.25 | 5.53±0.33 | 5.43 |
| L | 1 | 34.67 | 92.00 | -0.1927±0.3198 | 0.3180±0.1228 | 55.48±5.31 | 3.62±0.33 | 3.62 |
| | 2 | 16.67 | 81.33 | -0.3926±0.2284 | 0.1672±0.2411 | 68.70±7.65 | 2.88±0.63 | 2.87 |
| | 3 | 26.67 | 79.33 | -0.2886±0.2934 | 0.1639±0.2286 | 136.35±38.12 | 5.59±0.34 | 5.49 |

**Evaluation Metrics.**    We evaluate both methods using **Execution Success Rate (SR)**, **Average Robustness Value (RV)**, and **Total Planning Time (T0)** to assess the task satisfaction of the actually executed trajectories and the overall planning efficiency. Additionally, we record the **Trajectory Generation Time (T1)** to analyze the computational efficiency of the trajectory generation module of our algorithm. More details about these evaluation metrics and experiment settings are provided in **Appendix F.3**.

**Results and Analysis.**    Due to page limitations, we present a subset of the experimental results in Table 2, with the full results available in Table F.2. The results indicate that RGD works only on simple tasks (Types 1-3) and struggles with more complex tasks, as it cannot guarantee STL satisfaction and faces challenges balancing robustness optimization with dynamic feasibility for the entire long trajectories. In contrast, our method guarantees STL satisfaction and focuses the challenge on ensuring dynamic feasibility of short trajectory segments. In simpler environments, such as

U-Maze, our method achieves over **80%** execution success rate across all task types, and maintains at least **69%** success in the more complex Large environment. Moreover, the average robustness values of the actually executed trajectories using our method are consistently higher, further demonstrating its advantage in task satisfaction performance. Our approach is also more efficient, outperforming RGD by over $10\times$ in planning time. By comparing **Trajectory Generation Time (T1)** and **Total Planning Time (T0)**, we identify trajectory generation as the primary bottleneck in our algorithm. This is primarily due to the additional runtime introduced by `SafeDiffuser`, which uses complex quadratic programming for safety constraints enforcement. We plan to accelerate the trajectory generation module in future work by incorporating advanced sampling acceleration techniques for diffusion models [39, 40, 41].

### 5.2 Experiments under More Complex Dynamics

We further evaluate the generality of our framework on two dynamics-rich domains from OG-Bench [42]: Cube (6-DoF UR5e manipulator) and AntMaze (8-DoF Ant). Training uses only task-agnostic trajectory datasets provided by OGBench; at evaluation time we replace the original goal-conditioned objectives with STL specifications. In Cube we plan and track end-effector trajectories in Cartesian space, and in AntMaze we plan in the workspace and execute with a learned inverse-dynamics controller. Across nine STL templates per environment, the framework attains high execution success rate in Cube (over **84%**) and robust performance in AntMaze (over **60%**), with planning times that remain practical despite the increased dynamical complexity (Table F.3). These results indicate that the proposed decomposition-allocation-generation pipeline transfers beyond simple scenarios to higher-dimensional settings. Full experimental settings, metrics, and analyses are provided in **Appendix F.4**.

### 5.3 Comparative Experiment with Optimization-based Method

In the previous experiments, due to the lack of access to precise system dynamics in the simulation environment, we could not use sound-and-complete methods to verify the feasibility of randomly generated STL tasks. Instead, task feasibility was determined by our progress allocation module, which may lead to a slightly optimistic estimation of the overall success rate, as it overlooks the conservativeness of our approach. To more accurately assess this conservativeness, we conduct additional experiments in a custom-built environment with full access to both the environment and system dynamics. In this setting, we adopt a sound and complete optimization-based method [4], which utilizes full model information to rigorously verify the feasibility of STL formulas. Our algorithm is then evaluated on those verified feasible cases. Results show that the progress allocation module achieves a success rate exceeding **80%** across various task scenarios, highlighting its strong capability to identify feasible solutions. Full experimental details are provided in **Appendix F.5**.

## 6 Conclusion

We presented a data-driven hierarchical framework for planning trajectories that satisfy complex Signal Temporal Logic (STL) tasks under unknown dynamics using only task-agnostic offline trajectories. The approach semantically decomposes an STL formula into time-aware progresses, allocates timed waypoints via a dynamics-aware search guided by a learned reachability heuristic, and synthesizes short constrained trajectory segments with a diffusion-based generator; these segments are then stitched into complete trajectories. We established formal soundness of the planning pipeline for STL satisfaction, and validated the method extensively. On Maze2D, the framework attains high execution success rate on long-horizon tasks while delivering over $10\times$ speedups relative to a robustness-guided diffusion baseline and higher robustness values with smaller variance. On dynamics-rich domains (Cube and AntMaze), it maintains strong success rates with practical planning times, and in a custom environment with known dynamics the progress-allocation module achieves $>80\%$ feasibility on rigorously certified instances, underscoring completeness in practice. Overall, the results indicate that unifying logic-level task decomposition with data-driven trajectory synthesis yields a scalable and general solution for zero-shot STL planning; future work will pursue stronger time-prediction and control modules, verification enhancements, and accelerated diffusion sampling for even longer horizons.

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

# A    Related Works

## A.1    STL Decomposition

To address the high complexity of STL control synthesis problems, several decomposition-based methods have been proposed [43, 44, 26]. In [43], the authors proposed a formula transformation-based method for multi-agent STL planning. This approach jointly decomposes an STL specification and team of agents. In [44], the authors decompose STL tasks into several subtasks with non-overlapping time intervals using time interval decomposition and sequentially apply the shrinking horizon Model Predictive Control (MPC) algorithm to each short-time-interval subtask. However, this method is limited to handling STL fragments that do not include nested temporal logic operators.

The work most similar to our STL decomposition framework is [26]. This work first decomposes STL tasks into several spatio-temporal subtasks with time variable constraints. Then, through time variable simplification, partial ordering, and slicing, the subtasks are segmented into several time intervals. Finally, a planning algorithm is subsequently used to sequentially solve the atomic tasks within each time interval. Our algorithm similarly begins by decomposing the STL task into several spatio-temporal progresses and time variable constraints. However, we adopt a search-based approach to determine the completion times and corresponding states to achieve each progress. During the search process, our method dynamically maintains the time variable constraints on-the-fly according to the completion of progresses. By incorporating the search mechanism, our algorithm achieves greater completeness compared to the incremental planning approach used in [26]. Additionally, the dynamic maintenance of time variable constraints enables a more natural handling of relationships between subtasks and extends the applicability of our approach to more complex STL task fragments that allows "until" operator.

## A.2    Planning with Diffusion Model

Recent advancements in diffusion-based planning methods highlight their remarkable flexibility, as they rely exclusively on offline trajectory datasets and do not require direct interaction with or access to the environment. By leveraging guided sampling, these methods can address a wide range of objectives without the need for retraining. This approach has been widely applied to long-horizon task planning and decision-making, facilitating the generation of states or actions for control purposes [16, 17, 18].

In the domain of diffusion-based planning for temporal logic tasks, several significant studies have been conducted [23, 24, 21, 22]. For instance, [21] proposed a classifier-based guidance approach to direct the sampling process of the diffusion model, enabling the generation of trajectories that fulfill finite Linear Temporal Logic ($LTL_f$) tasks. Similarly, [22] introduced a data-driven hierarchical framework that decomposes co-safe LTL tasks into sub-tasks using hierarchical reinforcement learning. This framework integrates a diffusion model with a determinant-based sampling technique to efficiently produce diverse low-level trajectories, enhancing both planning success rates and task generalization capabilities.

For diffusion-based Signal Temporal Logic (STL) planning, [23] employed a classifier-based guidance method that leverages the gradient of robustness values to guide the sampling process of a diffusion model. This method enabled the generation of vehicle trajectories compliant with STL-specified traffic rules. Expanding on this work, [24] introduced a data augmentation process to further improve trajectory diversity and increase the satisfaction rate of specified rules. However, these approaches remain constrained to relatively simple STL tasks due to the inherent complexity of optimizing robustness values, as well as the trade-off between maximizing reward objectives and preserving the feasibility of generated trajectories [25].

We compare our algorithm with the method proposed in [23] in the experiments. The results demonstrate that our algorithm can handle complex and long-horizon STL tasks that the method in [23] cannot address, while also exhibiting significant advantages in runtime efficiency.

## A.3    Trajectory Stitching

In practical applications of diffusion-based trajectory planning, the training data for diffusion models typically consist of short and simple trajectories. As a result, the learned models struggle to capture the

distribution of more complex behaviors, making it difficult to generate trajectories that satisfy intricate task specifications through guided sampling. To address this limitation, recent studies [45, 46, 47, 48] leverage *trajectory stitching* methods. The core idea is to construct long-horizon trajectories by stitching together shorter trajectory segments, thereby enabling the satisfaction of more challenging tasks. Current research on trajectory stitching mainly focuses on data augmentation [45, 48] or enabling diffusion models, trained on simple sub-optimal trajectories, to perform relatively complex long-horizon goal-reaching tasks [46, 47].

In contrast to these works, we focus on using trajectory stitching to generate trajectories that satisfy complex, long-horizon STL tasks by leveraging a diffusion model trained on task-agnostic, simple trajectory data. We address the challenge of generating trajectories that satisfy complex STL specifications by decomposing the global task into simpler local sub-tasks using an hierarchical STL-decomposition and planning framework. For each sub-task, a diffusion model is employed to generate a locally feasible trajectory segment, and the final trajectory is obtained by stitching these segments together. The key difficulty lies in ensuring that the decomposition and stitching process preserves satisfaction of the original global STL specification.

# B    Limitations

## B.1    Algorithmic Incompleteness

The main limitation of the proposed framework lies in its *incompleteness*-the planner may fail to find a solution even when one exists-due to three inherent sources of conservativeness:

(i) **Specification strengthening.** Eliminating disjunctions transforms the original formula $\varphi$ into a logically stronger formula $\tilde{\varphi}$. While this may exclude some feasible solutions, such transformation is typical in STL planning [30].

(ii) **Heuristic allocation.** The depth-first, heuristic-based progress allocation (Algorithm 1) relies on approximate time-to-reach estimations and greedy ordering, which may prematurely prune feasible branches. This can be partially mitigated by treating the first feasible solution as a candidate rather than terminating immediately. Exploring more candidates improves solution quality but increases planning time.

(iii) **Data-driven synthesis.** The diffusion model can only generate trajectories that resemble its training distribution. When the task requires behaviors beyond this distribution, the model may fail to synthesize valid segments.

In fact, the incompleteness is *inevitable* in any sound, data-driven planner under this scenario. Achieving completeness would require full knowledge of the true system dynamics $f$, contradicting the core assumption of the problem. Therefore, pursuing absolute completeness is neither meaningful nor practical. Instead, a sound yet statistically conservative framework offers greater value in real-world applications.

We outline some directions that can *reduce* (though never eliminate) the conservativeness:

- **Adaptive progress allocation.** Replacing the greedy allocation with a beam search or widening strategy, and integrating uncertainty-aware arrival-time predictors (e.g., conformal prediction bounds), can diminish the risk of pruning viable branches.

- **Hybrid model refinement.** When limited model knowledge is available (e.g., a coarse dynamics model or learned residual), incorporating it as a constraint inside trajectory generator may shrink the realism gap and enhance synthesis coverage.

- **Data Augmentation for Synthesis Coverage.** Enhance the diffusion prior and Time Predictor using stitching-based data augmentation [45], which increases the coverage of rare transitions and long detours. This augmentation mitigates the sensitivity of data-driven synthesis to dataset sparsity and distribution shift.

We leave these possible improvements for future work.

## B.2 Optimality

Our current objective prioritizes *task correctness* and *dynamic feasibility* under unknown dynamics. The optimization of secondary criteria such as makespan, temporal slack, energy, or risk remains outside our present scope and is particularly challenging in our setting.

Nevertheless, incorporating optimality represents a promising research direction. A practical approach is to replace the current "first-feasible" strategy with an *anytime* allocator that continues to explore the search space after finding an initial solution, thereby trading additional computation for improved objectives such as reduced makespan or increased slack within invariance windows. This approach can be combined with (i) multi-objective scoring that integrates feasibility and soft cost terms, (ii) admissible pruning based on uncertainty-aware time-to-reach bounds, and (iii) post-allocation local refinements such as waypoint re-timing under STL windows while maintaining soundness.

It is important to emphasize that any extension toward optimality must still preserve dynamic feasibility. Overemphasizing cost minimization may increase dynamic infeasibility during downstream synthesis or execution. Systematically balancing these trade-offs through anytime search, Pareto-front maintenance, or constrained optimization over progress schedules is an important direction for future work.

## B.3 Scenario Limitations

As a planning framework, our method currently performs planning in a relatively low-dimensional task space rather than directly in the high-dimensional configuration space. During execution, control is still handled by a low-level controller or an learned inverse dynamics model, as described in Section 3.5. This design choice may limit the planner's ability to fully account for low-level dynamic feasibility when generating trajectories, resulting in a potential gap between the planned and the executed trajectories. Nevertheless, our experiments evaluate the success rate based on the actually executed trajectories, and the results show that even in challenging environments with complex low-level dynamics, such as AntMaze, our method still achieves satisfactory performance.

In addition, our current experiments primarily focus on *navigation-style* STL tasks, in which the atomic predicates typically specify reaching certain regions, and the overall task requires sequentially visiting designated goals within specified time windows. This follows the mainstream setting adopted in most studies on temporal logic task planning and control synthesis [12, 21, 22, 30, 49, 26]. However, temporal logic can also describe other types of behaviors, such as safety constraints and maintenance conditions [24, 50]. These scenarios usually involve relatively simple temporal logic specifications, whereas our focus is on planning under long-horizon and complex temporal logic tasks. Therefore, we restrict our experiments to navigation-style tasks and leave other types of STL scenarios for future exploration.

## B.4 Time Predictor Limitations

The Time Predictor (TP) uses offline trajectories to estimate time-to-reach between states, thereby injecting a notion of dynamical feasibility into allocation. This data-driven design has inherent limitations. When datasets are sparse or misaligned with the evaluation distribution, the TP can overfit local transition patterns or extrapolate poorly to pairs of states that do not appear in the data. This issue becomes more pronounced when avoidance constraints require long detours. Coverage requirements also increase with the dimensionality of the planning space, and without sufficient cross-episode transitions the predictions can become biased or insufficiently dispersed, which may lead to unreachable schedules.

**Effectiveness within our regime.** In the data regimes considered here, where abundant task-agnostic trajectories are available, the TP works in concert with the generator and the controller. The TP proposes a horizon, the diffusion-based generator synthesizes a segment of the corresponding length, and the time-synchronous controller executes it. The predictor-generator-controller pipeline achieves high execution reliability in practice (as shown in Section F.6). These results indicate that moderate TP accuracy is often sufficient when combined with segment-level generation and closed-loop tracking in our framework.

**Mitigations and modular improvements.** Within our modular framework, several upgrades can reduce TP-induced failures while preserving soundness:

- **Stronger time-prediction models.** Replace the lightweight TP with more expressive predictors that can better capture long detours and heterogeneous dynamics. Options include models with calibrated uncertainty as well as recent approaches that learn temporal-distance or hitting-time style quantities [51, 52, 53]. These predictors can be paired with slack-aware scheduling to translate predictions into more robust waypoint times.
- **Data augmentation for coverage.** Increase support for rare transitions through stitching-based augmentation [45], which composes cross-episode segments to create longer feasible paths and improves coverage in sparsely sampled regions.

We leave these possible improvements for future work.

## C  Semantics of Signal Temporal Logic

### C.1  Boolean Semantics

The Boolean semantics of a STL formula $\varphi$ with respect to a signal $\mathbf{s}_t$ starting at time $t$ are defined inductively as follows [54]:

$$\mathbf{s}_t \vDash \mu \Leftrightarrow h_\mu(\mathbf{x}_t) \geq 0, \tag{C.1}$$

$$\mathbf{s}_t \vDash \varphi_1 \wedge \varphi_2 \Leftrightarrow \mathbf{s}_t \vDash \varphi_1 \wedge \mathbf{s}_t \vDash \varphi_2, \tag{C.2}$$

$$\mathbf{s}_t \vDash \varphi_1 \vee \varphi_2 \Leftrightarrow \mathbf{s}_t \vDash \varphi_1 \vee \mathbf{s}_t \vDash \varphi_2, \tag{C.3}$$

$$\mathbf{s}_t \vDash \mathrm{F}_{[a,b]}\varphi \Leftrightarrow \exists t' \in [t+a, t+b] \text{ s.t. } \mathbf{s}_{t'} \vDash \varphi, \tag{C.4}$$

$$\mathbf{s}_t \vDash \mathrm{G}_{[a,b]}\varphi \Leftrightarrow \forall t' \in [t+a, t+b] \text{ s.t. } \mathbf{s}_{t'} \vDash \varphi, \tag{C.5}$$

$$\mathbf{s}_t \vDash \varphi_1 \mathrm{U}_{[a,b]}\varphi_2 \Leftrightarrow \exists t' \in [t+a, t+b] \text{ s.t. } \mathbf{s}_{t'} \vDash \varphi_2$$
$$\wedge \, \forall t'' \in [t, t'], \mathbf{s}_{t''} \vDash \varphi_1. \tag{C.6}$$

For simplicity, we slightly abuse the notation $\mathbf{x}_t \vDash \mu$ to denote $h_\mu(\mathbf{x}_t) \geq 0$, and thus, we have

$$\mathbf{s}_t \vDash \mu \Leftrightarrow h_\mu(\mathbf{x}_t) \geq 0 \Leftrightarrow \mathbf{x}_t \vDash \mu. \tag{C.7}$$

### C.2  Quantitative Semantics

Besides the Boolean semantics, STL also incorporates a concept of robustness value [54], which refers to its quantitative semantics used to measure the degree to which a signal satisfies or violates a formula. Positive robustness values signify satisfaction, while negative values indicate violation. The quantitative semantics of a STL formula $\varphi$ with respect to a signal $\mathbf{s}_t$ starting at time $t$ are defined as follows:

$$\rho(\mathbf{s}_t, \top) = \rho_{\max}, \text{ where } \rho_{\max} > 0,$$
$$\rho(\mathbf{s}_t, \mu) = h_\mu(\mathbf{x}_t),$$
$$\rho(\mathbf{s}_t, \neg\mu) = -h_\mu(\mathbf{x}_t),$$
$$\rho(\mathbf{s}_t, \varphi_1 \wedge \varphi_2) = \min(\rho(\mathbf{s}_t, \varphi_1), \rho(\mathbf{s}_t, \varphi_2)),$$
$$\rho(\mathbf{s}_t, \varphi_1 \vee \varphi_2) = \max(\rho(\mathbf{s}_t, \varphi_1), \rho(\mathbf{s}_t, \varphi_2)),$$
$$\rho(\mathbf{s}_t, \mathrm{F}_{[a,b]}\varphi) = \max_{t' \in [t+a, t+b]} \rho(\mathbf{s}_{t'}, \varphi), \tag{C.8}$$
$$\rho(\mathbf{s}_t, \mathrm{G}_{[a,b]}\varphi) = \min_{t' \in [t+a, t+b]} \rho(\mathbf{s}_{t'}, \varphi),$$
$$\rho(\mathbf{s}_t, \varphi_1 \mathrm{U}_{[a,b]}\varphi_2) = \max_{t' \in [t+a, t+b]} \min\{\rho(\mathbf{s}_{t'}, \varphi_2),$$
$$\min_{\tau \in [t, t']} \rho(\mathbf{s}_\tau, \varphi_1)\}.$$

## D  Semantic-Based STL Decomposition

The decomposition procedure is defined recursively as follows:

- If $\varphi = \mathrm{F}_{[a,b]}\mu$, then we have $\mathbb{P}_\varphi = \{\mathcal{R}(\lambda_i, \lambda_i, \mu)\}$ and $\mathbb{T}_\varphi = \{\lambda_i \in [a,b]\}$, $\Lambda_\varphi = \{\lambda_i\}$, where $\lambda_i$ is a new time variable.

- If $\varphi = \mathrm{G}_{[a,b]}\mu$, then we have $\mathbb{P}_\varphi = \{\mathcal{I}(a,b,\mu)\}$ and $\mathbb{T}_\varphi = \varnothing$, $\Lambda_\varphi = \varnothing$.

- If $\varphi = \mu_1 \mathrm{U}_{[a,b]}\mu_2$, then we have $\mathbb{P}_\varphi = \{\mathcal{I}(0, \lambda_i, \mu_1), \mathcal{R}(\lambda_i, \lambda_i, \mu_2)\}$ and $\mathbb{T}_\varphi = \{\lambda_i \in [a,b]\}$, $\Lambda_\varphi = \{\lambda_i\}$, where $\lambda_i$ is a new time variable.

- If $\varphi = \varphi_1 \wedge \varphi_2$, then we merge the progresses and time constraints, i.e., $\mathbb{P}_\varphi = \mathbb{P}_{\varphi_1} \uplus \mathbb{P}_{\varphi_2}$ and $\mathbb{T}_\varphi = \mathbb{T}_{\varphi_1} \uplus \mathbb{T}_{\varphi_2}$, $\Lambda_\varphi = \Lambda_{\varphi_1} \uplus \Lambda_{\varphi_2}$. We use "$\uplus$" to emphasize that the two merged sets contain no duplicate elements.

- If $\varphi' = \mathrm{F}_{[a,b]}\varphi$, then we (i) introduce a new time variable $\lambda_i$, i.e., $\Lambda_{\varphi'} = \Lambda_\varphi \uplus \{\lambda_i\}$ ; (ii) add a new time constraint $\mathbb{T}_{\varphi'} = \mathbb{T}_\varphi \uplus \{\lambda_i \in [a,b]\}$; (iii) increase each time indices in each progress by $\lambda_i$, i.e., $\mathbb{P}_{\varphi'} = \{\mathcal{P}(c_\Lambda + \lambda_i, d_\Lambda + \lambda_i, \mu) \mid \mathcal{P}(c_\Lambda, d_\Lambda, \mu) \in \mathbb{P}_\varphi\}$.
  Specifically, if $\varphi' = \mathrm{F}_{[a,a]}\varphi$, then we do not introduce a new time variable. In this case, we set $\Lambda_{\varphi'} = \Lambda_\varphi, \mathbb{T}_{\varphi'} = \mathbb{T}_\varphi, \mathbb{P}_{\varphi'} = \{\mathcal{P}(c_\Lambda + a, d_\Lambda + a, \mu) \mid \mathcal{P}(c_\Lambda, d_\Lambda, \mu) \in \mathbb{P}_\varphi\}$.

- If $\varphi' = \mathrm{G}_{[a,b]}\varphi$, we treat it as $\varphi' = \bigwedge_{k \in [a,b]} \mathrm{F}_{[k,k]}\varphi$. Each subformula $\mathrm{F}_{[k,k]}\varphi$ corresponds to a separate copy of progresses $\mathbb{P}_\varphi^{(k)}$, time variables $\Lambda_\varphi^{(k)}$ and the associated constraints $\mathbb{T}_\varphi^{(k)}$. Therefore, $\Lambda_{\varphi'} = \biguplus_{k \in [a,b]} \Lambda_\varphi^{(k)}, \mathbb{T}_{\varphi'} = \biguplus_{k \in [a,b]} \mathbb{T}_\varphi^{(k)}, \mathbb{P}_{\varphi'} = \bigcup_{k \in [a,b]} \{\mathcal{P}(c_\Lambda + k, d_\Lambda + k, \mu) \mid \mathcal{P}(c_\Lambda, d_\Lambda, \mu) \in \mathbb{P}_\varphi^{(k)}\}$.
  Specifically, for an invariance progress in $\mathbb{P}_\varphi$ of the form $\mathcal{I}(c, d, \mu)$ that does not contain any time variables, applying the operation $\mathrm{F}_{[k,k]}\varphi$ on its copy in $\mathbb{P}_\varphi^{(k)}$ yields $\mathcal{I}(c + k, d + k, \mu)$. We then merge all such copies $\mathcal{I}(c + k, d + k, \mu)$ for $k \in [a,b]$ into a single progress $\mathcal{I}(c + a, d + b, \mu)$.

- If $\varphi' = \phi \mathrm{U}_{[a,b]}\varphi$, then (i) introduce a new time variable $\lambda_i$, i.e., $\Lambda_{\varphi'} = \Lambda_\varphi \uplus \{\lambda_i\}$; and (ii) add a new time constraint, i.e., $\mathbb{T}_{\varphi'} = \mathbb{T}_\varphi \uplus \{\lambda_i \in [a,b]\}$; and (iii) increase each time indices in each progress in $\mathbb{P}_\varphi$ by $\lambda_i$ and modify each invariance progress $\mathcal{I}(c, d, \mu) \in \mathbb{P}_\phi$ to $\mathcal{I}(c, d + \lambda_i, \mu)$, i.e., $\mathbb{P}_{\varphi'} = \{\mathcal{P}(c_\Lambda + \lambda_i, d_\Lambda + \lambda_i, \mu) \mid \mathcal{P}(c_\Lambda, d_\Lambda, \mu) \in \mathbb{P}_\varphi\} \cup \{\mathcal{I}(c, d + \lambda_i, \mu) \mid \mathcal{I}(c, d, \mu) \in \mathbb{P}_\phi\}$.

**Remark 2.** *Since we require that, for any subformula of the form $\varphi_1 \mathrm{U}_{[a,b]}\varphi_2$, the subformula $\varphi_1$ does not contain the operators $\mathrm{F}$ or $\mathrm{U}$, it follows that in the case of $\varphi' = \phi \mathrm{U}_{[a,b]}\varphi$, no time variables or reachability progresses are introduced by the operations within $\phi$. Therefore, we have $\Lambda_\phi = \varnothing$, $\mathbb{T}_\phi = \varnothing$, and $\mathbb{P}_\phi = \{\mathcal{I}(c, d, \mu)\}$.*

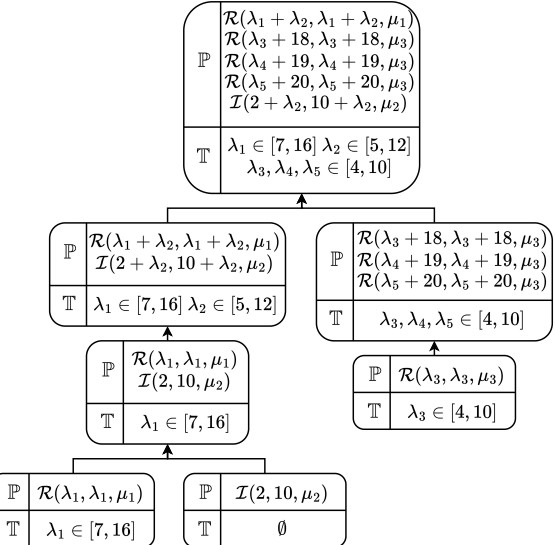

Figure D.1: Decomposition Process of STL Formula (D.1).

**Example 1.** *To illustrate the above progress decomposition process, we consider the following STL formula*

$$\varphi = \mathrm{F}_{[5,12]}\big(\mathrm{F}_{[7,16]}\mu_1 \wedge \mathrm{G}_{[2,10]}\mu_2\big) \wedge \mathrm{G}_{[18,20]}\mathrm{F}_{[4,10]}\mu_3. \tag{D.1}$$

*The decomposition process is shown in Figure D.1, where the progress and time constraints are constructed incrementally from the bottom to the top. The top node represents the overall decomposed $(\mathbb{P}_\varphi, \mathbb{T}_\varphi)$ for the STL formula $\varphi$.*

# E   Trajectory Generation

## E.1   Diffusion Models for Trajectory Planning

The core concept of diffusion-model-based trajectory planning[16, 17] is to employ the diffusion model to learn the distribution of pre-collected trajectories $q\left(\tau^0\right)$ in the current environment under the system dynamics, thereby transforming planning or control synthesis problems into conditional trajectory generation.

Diffusion models consist of two processes: the **diffusion process** and the **denoising process**.

The **diffusion process** gradually adds Gaussian noise to a trajectory $\tau^0$, transforming it into noise. At each timestep $i$, the noisy trajectory is given by:

$$q(\tau^i \mid \tau^{i-1}) = \mathcal{N}(\tau^i; \sqrt{1 - \beta_i}\tau^{i-1}, \beta_i\mathbf{I}), \tag{E.1}$$

where $\beta_i$ controls the noise scale, and $N$ is the total number of diffusion steps. The noisy trajectory $\tau^i$ at any step can be directly computed as:

$$\tau^i = \sqrt{\bar{\alpha}_i}\tau^0 + \sqrt{1 - \bar{\alpha}_i}\varepsilon, \quad \varepsilon \sim \mathcal{N}(\mathbf{0}, \mathbf{I}), \tag{E.2}$$

with $\bar{\alpha}_i = \prod_{j=1}^i (1 - \beta_j)$.

The **denoising process** reverses the diffusion by iteratively recovering the trajectory from Gaussian noise. The reverse distribution is approximated as:

$$p_\theta(\tau^{i-1} \mid \tau^i) = \mathcal{N}(\tau^{i-1}; \boldsymbol{\mu}_\theta(\tau^i, i), \boldsymbol{\Sigma}^i), \tag{E.3}$$

where $\boldsymbol{\mu}_\theta(\tau^i, i)$ is parameterized by the model, and $\boldsymbol{\Sigma}^i$ is typically fixed. Instead of learning $\boldsymbol{\mu}_\theta$ directly, the model typically predicts the noise $\varepsilon_\theta(\tau^i, i)$ and gets $\boldsymbol{\mu}_\theta(\tau^i, i)$ according to the relationship[15]:

$$\boldsymbol{\mu}_\theta(\tau^i, i) = \frac{1}{\sqrt{\alpha_i}}\left(\tau^i - \frac{1 - \alpha_i}{\sqrt{1 - \bar{\alpha}_i}}\varepsilon_\theta(\tau^i, i)\right). \tag{E.4}$$

The model is trained by minimizing a simplified loss function:

$$\mathcal{L}(\boldsymbol{\theta}) = \mathbb{E}_{i, \varepsilon, \tau^0}\left[\left\|\varepsilon - \varepsilon_\theta(\tau^i, i)\right\|_2^2\right]. \tag{E.5}$$

In this paper, superscripts denote the diffusion time step, while subscripts indicate the trajectory time step. For example, $\tau_t^i$ represents the state at trajectory time step $t$ during diffusion time step $i$. For noise-free trajectories, $\tau^0$, we omit the superscript when there is no ambiguity.

## E.2   Trajectory Stitching

Directly generating an entire long-horizon trajectory that meets all STL constraints is challenging, since the resulting trajectories may fall far outside the training distribution, which typically consists of shorter demonstrations. Inspired by recent work on trajectory stitching [45, 47], we decompose the problem into segment-wise generation. Specifically, between two adjacent way-points $(\mathbf{x}_i, t_i)$ and $(\mathbf{x}_{i+1}, t_{i+1})$, we generate a segment $\tau_i$ that:

1. has length $t_{i+1} - t_i + 1$;
2. starts at $\mathbf{x}_i$ and ends at $\mathbf{x}_{i+1}$;
3. satisfies all invariance progresses active in $[t_i, t_{i+1}]$.

The complete trajectory $\tau$ is then formed by concatenating all segments $\tau_i$. This design balances tractability with expressiveness, ensuring that long-horizon STL tasks can be addressed while maintaining consistency with the learned trajectory distribution.

### E.3 Constrained Generation Mechanisms

To enforce the above conditions, we incorporate several mechanisms into the generation process:

- **Length control.** As noted in [16], the planning horizon of a diffusion model is not fixed by its architecture but dynamically adapts based on the input noise size. This allows for variable-length plans during the denoising process at test time. In this work, we leverage this adaptability by directly adjusting the input noise size to generate trajectories of the desired length. During experiments, we observe that models trained with longer planning horizons generalize better to different trajectory lengths during testing. In contrast, models trained with shorter planning horizons exhibit weaker generalization. To address this, one can employ multiple models trained with different horizons, each responsible for generating trajectories whose lengths are close to their respective training horizons. Alternatively, one can train one diffusion model using trajectory segments of varying lengths, which has been shown to improve the generalization ability of diffusion models for generating trajectories of different lengths [53].

- **Waypoint inpainting.** To respect waypoint constraints, we treat generation as an inpainting problem as [16], where the start and end states of each segment are replaced with $(\mathbf{x}_i, \mathbf{x}_{i+1})$ after every denoising step.

- **Safety constraints.** Each invariance progress $\mathcal{I}(a, b, \mu)$ is decomposed into constraints of the form $h_{\mu_i}(\mathbf{x}_t) \geq 0$ for $t = a, a+1, \ldots, b$, which can be handled by many existing methods [29, 32, 33, 34, 35]. We adopt `SafeDiffuser` [29], which integrates control barrier functions (CBFs) [36] into the denoising dynamics. At each diffusion step, a quadratic program (QP) is solved to minimally adjust the update direction while ensuring all CBF constraints are satisfied, thereby providing finite-time invariance guarantees during trajectory generation.

## F Details of Experiments

### F.1 Environments

We evaluate our proposed framework across a diverse set of environments that encompass both navigation and robotic manipulation domains, as well as a custom-built environment designed for controlled comparisons. These environments differ significantly in their underlying dynamics and control dimensionality, thereby providing a comprehensive test of the generality and robustness of our approach. Importantly, in all experiments we only utilize the environments and their associated *STL task-agnostic* trajectory datasets provided by the original benchmarks. We explicitly discard the original task definitions and evaluation protocols, and instead redefine all tasks as Signal Temporal Logic (STL) specifications, allowing us to unify the evaluation across domains. Visualizations of all environments are shown in Figure F.1.

- **D4RL Maze2D [38] (*U-Maze*, *Medium*, *Large*).** This environment features a point-mass agent navigating within two-dimensional maze layouts of increasing complexity. The original benchmark tasks are goal-reaching problems defined by start and goal locations. In our experiments, these tasks are reformulated as STL specifications that require the agent to sequentially reach multiple target regions in the maze, enforcing both temporal ordering and spatial reachability constraints.

- **OGBench AntMaze [42].** In this environment, an 8-DoF quadruped "ant" robot must traverse a large two-dimensional maze. Compared to Maze2D, the AntMaze environment introduces substantially more complex locomotion dynamics. While the original benchmark evaluates performance by whether the ant reaches a single goal region, we instead define STL tasks that require visiting multiple spatial regions in a prescribed temporal order.

- **OGBench Cube [42].** This environment involves a 6-DoF UR5e robotic arm performing cube manipulation on a tabletop. The original benchmark tasks assess goal-conditioned manipulation success using predefined demonstrations. For our evaluation, we abstract away from the physical cubes and focus on the Cartesian-space motion of the robot's end-effector. We define STL tasks that constrain the end-effector to visit a sequence of spatial waypoints within specified temporal windows, thereby converting the manipulation process into a structured STL-constrained trajectory planning problem. This abstraction allows large-scale and systematic testing of our framework.

- **Custom-built Environment**. To further evaluate the reliability of the Progress Allocation module in our framework, we construct a custom simulation environment consisting of a bounded two-

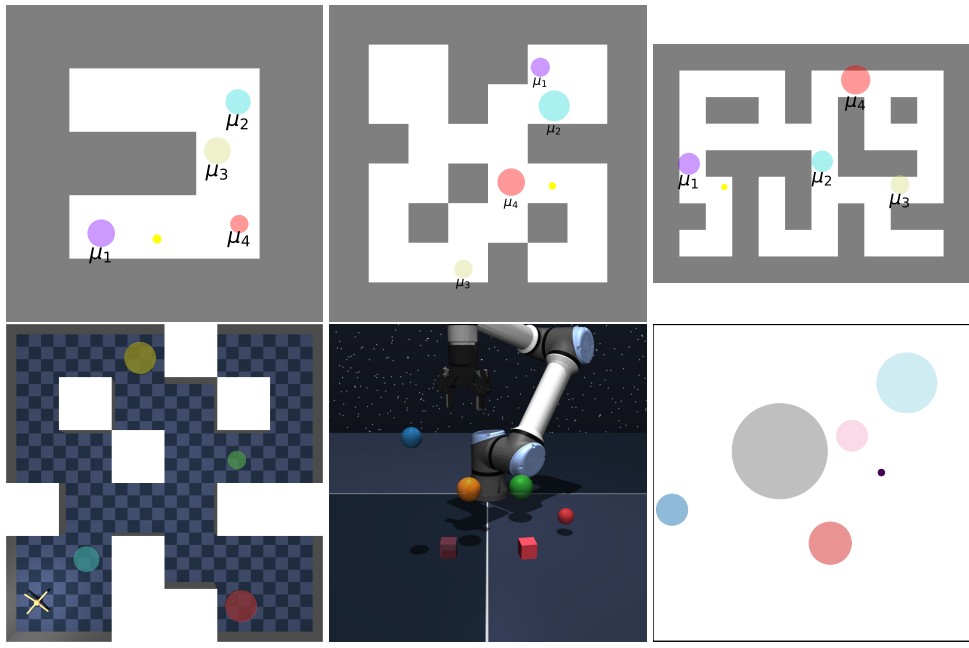

Figure F.1: Visualization of the environments used in our experiments. From left to right and top to bottom: the three Maze2D environments (*U-Maze*, *Medium*, *Large*), the AntMaze environment, the Cube environment and the custom-built environment.

dimensional square arena with a circular obstacle. The agent follows double-integrator dynamics and is required to accomplish randomly generated STL tasks by reaching designated regions within specified time intervals. Unlike the benchmark environments, this setting provides full access to the system model and environment information, which are relatively simple, thereby enabling direct comparison with sound and complete optimization-based baselines that require full knowledge of system dynamics and environment layout.

## F.2 STL Task Generation

To generate random STL tasks, we design nine STL task templates, as illustrated in Table F.1. For each template, we randomly sample time intervals as well as the positions and sizes of the regions corresponding to the atomic predicates, thereby producing diverse randomized STL tasks. The feasibility of each generated STL task is verified using the Progress Allocation module of our method in all experiments, except for the custom-built environment experiment, where feasibility is instead verified using the sound-and-complete baseline algorithm.

Table F.1: STL Task Templates for Experiments

| Type | STL Templates |
|------|---------------|
| 1 | $F_{I_1}\mu_1 \wedge G(\neg\mu_2)$ |
| 2 | $F_{I_1}\mu_1 \wedge F_{I_2}\mu_2$ |
| 3 | $F_{I_1}\mu_1 \wedge (\neg\mu_1 U_{I_1}\mu_2)$ |
| 4 | $F_{I_1}(\mu_1 \wedge (F_{I_2}(\mu_2 \wedge F_{I_3}(\mu_3 \wedge F_{I_4}(\mu_4)))))$ |
| 5 | $F_{I_1}(\mu_1 \wedge (F_{I_2}(\mu_2 \wedge F_{I_3}(\mu_3)))) \wedge G(\neg\mu_4)$ |
| 6 | $F_{I_1}(\mu_1) \wedge F_{I_2}(\mu_2) \wedge F_{I_3}(\mu_3) \wedge G(\neg\mu_4)$ |
| 7 | $F_{I_1}(G_{I_2}(\mu_1)) \wedge F_{I_3}(\mu_2) \wedge G(\neg\mu_3)$ |
| 8 | $F_{I_1}(\mu_1 \wedge F_{I_2}(G_{I_3}(\mu_2)))$ |
| 9 | $F_{I_1}(\mu_1 \wedge F_{I_2}(\mu_2) \wedge F_{I_3}(\mu_3) \wedge G_{I_4}(\mu_4))$ |

## F.3 Details of Experiment in Maze2d Environment

**Baseline Algorithm.** We compare our algorithm with the method proposed in [23], which adopts classifier-based guidance and directly leverages the gradient of the trajectory's robustness value to guide the sampling process of the diffusion model, thereby optimizing the robustness of the generated trajectory. The gradient of robustness is calculated by the STLCG method proposed in [55]. In the following text, we refer to this algorithm as the Robustness Guided Diffuser (RGD).

**Experimental Settings.** The diffusion models used in both RGD and our algorithm are trained following the procedure in [16] using the D4RL dataset [38]. A simple multilayer perceptron (MLP) with four fully connected layers is used as the `TimePredict` model in our algorithm and it is also trained using the same dataset. In our experiments, we employ diffusion model to generate only the state sequence of the trajectory and use a simple PD controller to follow the state sequence during running to get the actual execution trajectory as described in Section 3.5.

**Evaluation Metrics.** For each pair of Maze2D environment (U-Maze, Medium, Large) and task template listed in Table F.1, we generate 150 feasible random STL formulae as testing cases and test RGD and our algorithm on them and record the following metrics:

- **Execution Success Rate (SR):** The proportion of cases where the actual execution trajectory achieve non-negative robustness values.
- **Average Robustness Value (RV):** The average robustness value of the executed trajectories, after discarding the top and bottom 5% to mitigate the effect of outliers.
- **Total Planning Time (T0):** The average total running time (in seconds) to plan a trajectory per case.

In addition, we also record the average **Trajectory Generation Time (T1)**, which is the average time spent by the Trajectory Generation module of our algorithm per case. By recording this metric, we analyze the proportion of runtime contributed by each module in our algorithm.

For total planning time, we report both the mean and standard deviation in our results. For success rate, we compute the proportion of successful cases over the total number of test cases.

**Full Results.** The full experimental results are presented in Table F.2.

**More Cases.** We visualized some of the experimental results. The actual execution trajectories for some successful cases (where the STL tasks were satisfied by the execution trajectories) are shown in Figure F.2 to Figure F.4. For some failed cases (where the STL tasks were not satisfied), the trajectories planned by our algorithm are shown in Figure F.5.

**Failure-case Analysis.** Notably, by analyzing the failure-cases, we identified that the primary reason for execution failure is that the trajectories generated by the diffusion model significantly violated system dynamics, such as colliding with obstacles in the environment or having excessively large distances between consecutive states. To further enhance the actual execution success rate, our method can be integrated with some receding horizon control methods [56] or online replanning strategies [57]. This extension will be explored in our future work.

## F.4 Details of Experiments under More Complex Dynamics

To further evaluate the generality and robustness of our framework, we conduct experiments on two dynamics-rich domains from the offline goal-conditioned RL benchmark OGBench [42]: *cube-single-play* ("Cube") and *antmaze-medium-navigate* ("AntMaze").

**Experimental Settings.** The Cube environment involves a 6-DoF UR5e robotic arm manipulating a cube, while AntMaze features an 8-DoF quadruped ant navigating through a complex maze. In both environments, training relies solely on STL task-agnostic trajectory datasets provided by the benchmark; during evaluation, we replace the original goal-conditioned objectives with randomly generated STL tasks that encode multi-stage temporal and spatial constraints.

For AntMaze, we adopt the STL formulation described in Section 5.1: the agent must visit designated regions in a specific temporal order to satisfy the task specification. Planning is performed in the two-dimensional $x$-$y$ workspace, while execution uses an inverse dynamics controller that maps a 29-dimensional observation and the next $x$-$y$ target to an 8-dimensional action at each control step as

Table F.2: Full Result of Experiment in Maze2D Environment. U:U-Maze; M:Medium; L:Large; RGD: Robustness Guided Diffuser; T1:Trajectory Generation Time. "-" means that RGD fails to generate feasible trajectory.

| Env | Type | Success Rate(%)↑ | | Robustness Value↑ | | Total Planning Time(s)↓ | | T1(s) |
|---|---|---|---|---|---|---|---|---|
| | | RGD | ours | RGD | ours | RGD | ours | |
| U | 1 | 80.00 | 97.33 | 0.1084±0.1132 | 0.1938±0.0715 | 13.43±1.51 | 0.86±0.13 | 0.86 |
| | 2 | 36.67 | 92.00 | -0.2208±0.2826 | 0.1504±0.0814 | 16.65±2.06 | 0.64±0.17 | 0.64 |
| | 3 | 32.00 | 91.33 | -0.1695±0.2521 | 0.1354±0.0745 | 19.68±11.76 | 1.31±0.15 | 1.21 |
| | 4 | - | 90.00 | - | 0.1120±0.1424 | - | 1.64±0.20 | 1.38 |
| | 5 | - | 84.67 | - | 0.0921±0.0984 | - | 2.59±0.45 | 2.46 |
| | 6 | - | 86.67 | - | 0.1003±0.0806 | - | 2.35±0.42 | 2.35 |
| | 7 | - | 89.33 | - | 0.1293±0.0766 | - | 1.86±0.33 | 1.86 |
| | 8 | - | 97.33 | - | 0.1965±0.0435 | - | 0.98±0.15 | 0.81 |
| | 9 | - | 88.67 | - | 0.1047±0.1081 | - | 1.53±0.27 | 1.52 |
| M | 1 | 70.00 | 94.67 | 0.0885±0.2768 | 0.3205±0.0957 | 53.90±5.78 | 3.74±0.34 | 3.74 |
| | 2 | 34.67 | 89.33 | -0.2277±0.3089 | 0.2013±0.1950 | 70.69±8.27 | 2.72±0.67 | 2.72 |
| | 3 | 35.33 | 83.33 | -0.2393±0.3130 | 0.1761±0.2060 | 129.87±27.25 | 5.53±0.33 | 5.43 |
| | 4 | - | 83.33 | - | 0.1534±0.2615 | - | 7.09±0.54 | 6.80 |
| | 5 | - | 82.00 | - | 0.1457±0.2182 | - | 11.36±1.32 | 11.22 |
| | 6 | - | 84.67 | - | 0.1477±0.1977 | - | 11.76±1.36 | 11.76 |
| | 7 | - | 90.00 | - | 0.2148±0.1599 | - | 8.01±1.21 | 8.00 |
| | 8 | - | 91.33 | - | 0.2712±0.1772 | - | 3.87±0.38 | 3.71 |
| | 9 | - | 82.67 | - | 0.1320±0.2454 | - | 6.53±0.74 | 6.52 |
| L | 1 | 34.67 | 92.00 | -0.1927±0.3198 | 0.3180±0.1228 | 55.48±5.31 | 3.62±0.33 | 3.62 |
| | 2 | 16.67 | 81.33 | -0.3926±0.2284 | 0.1672±0.2411 | 68.70±7.65 | 2.88±0.63 | 2.87 |
| | 3 | 26.67 | 79.33 | -0.2886±0.2934 | 0.1639±0.2286 | 136.35±38.12 | 5.59±0.34 | 5.49 |
| | 4 | - | 69.33 | - | 0.0589±0.3133 | - | 7.46±0.52 | 7.13 |
| | 5 | - | 79.33 | - | 0.1324±0.2663 | - | 12.62±0.75 | 12.45 |
| | 6 | - | 73.33 | - | 0.1104±0.2673 | - | 12.37±1.35 | 12.37 |
| | 7 | - | 84.00 | - | 0.1770±0.2360 | - | 8.18±0.68 | 8.18 |
| | 8 | - | 85.33 | - | 0.2424±0.2502 | - | 3.93±0.33 | 3.75 |
| | 9 | - | 76.67 | - | 0.0823±0.2703 | - | 6.93±0.60 | 6.92 |

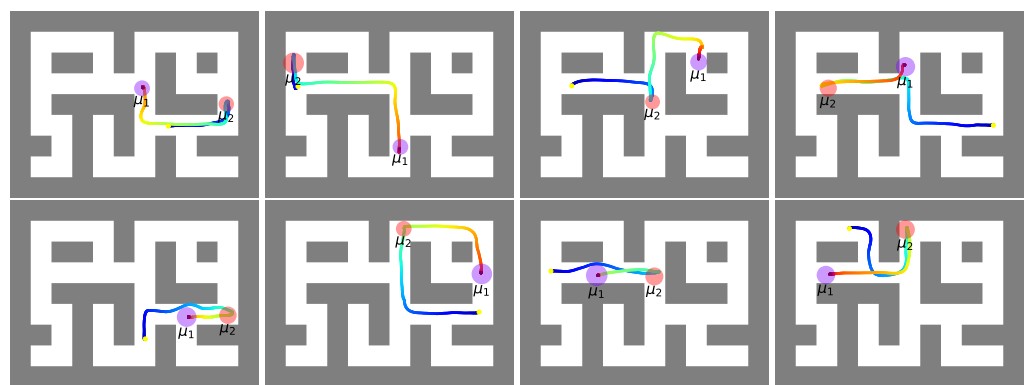

Figure F.2: Successful Execution Trajectories in Maze2D-large Environment under STL Template 3: $F_{I_1}\mu_1 \wedge (\neg\mu_1 U_{I_1}\mu_2)$. The task requires the agent to eventually reach region $\mu_1$ within a given time interval, but before reaching $\mu_1$, it must first visit region $\mu_2$.

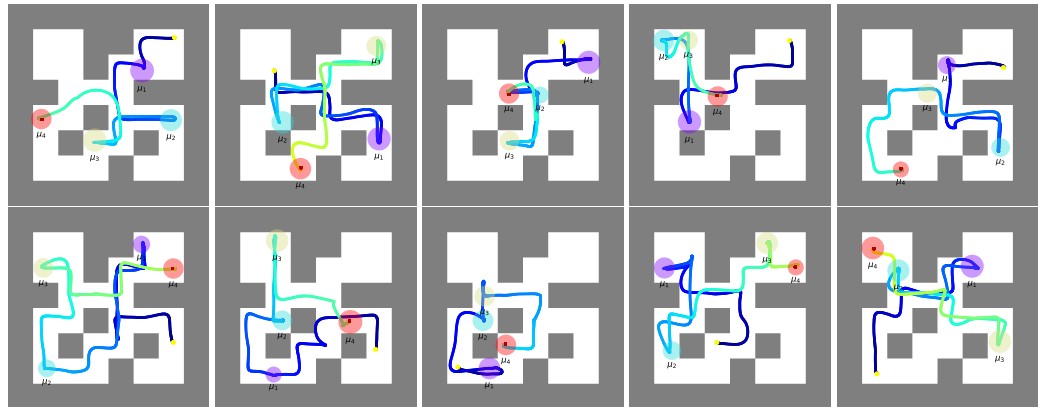

Figure F.3: Successful Execution Trajectories in Maze2D-medium Environment under STL Template 4: $F_{I_1}(\mu_1 \wedge (F_{I_2}(\mu_2 \wedge F_{I_3}(\mu_3 \wedge F_{I_4}(\mu_4)))))$. The task requires the agent to sequentially visit regions $\mu_1, \mu_2, \mu_3, \mu_4$ within the given time intervals.

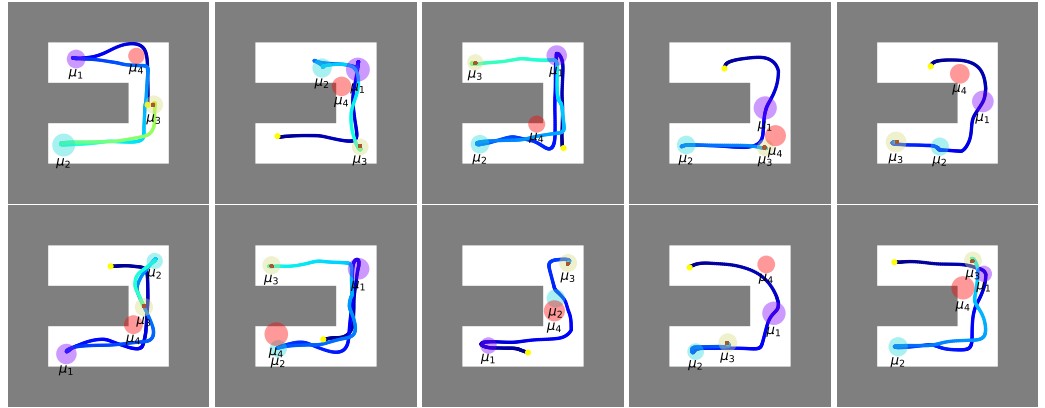

Figure F.4: Successful Execution Trajectories in Maze2D-umaze Environment under STL Template 5: $F_{I_1}(\mu_1 \wedge (F_{I_2}(\mu_2 \wedge F_{I_3}(\mu_3)))) \wedge G(\neg\mu_4)$. The task requires the agent to sequentially visit regions $\mu_1, \mu_2, \mu_3$ within the given time intervals and always avoid the region $\mu_4$.

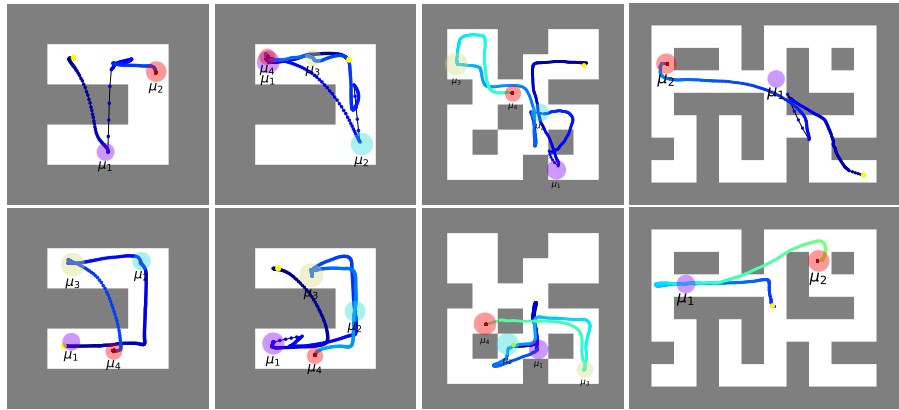

Figure F.5: Planned Trajectories in Some Failure Cases.

in [47]. Given the higher dynamic complexity of this domain, we set $k = 2$, such that each planning timestep corresponds to two control steps, as described in Section 3.5.

For the Cube environment, we focus on the end-effector's Cartesian motion rather than the physical manipulation of objects. Trajectories are generated in $x$-$y$-$z$ space and tracked via a PD controller, as in Section 3.5. This abstraction ensures methodological consistency with other experiments while still capturing the essential planning characteristics of high-dimensional robotic control.

**Evaluation Metrics.** For each environment and STL task template (as defined in Table F.1), we randomly generate 100 feasible STL formulae as test cases. We report two quantitative metrics: **Execution Success Rate (SR)** and **Total Planning Time (T0)**.

**Results.** The complete results are summarized in Table F.3. Our framework achieves high success rates across both environments, with moderate computational cost despite the distinct dynamic structures of the two systems.

Table F.3: Results in Cube and AntMaze. SR: Execution Success Rate; T0: Total Planning Time;

| Env | Type | SR(%) | T0(s) |
|-----|------|-------|-------|
| Cube | 1 | 100.0 | 0.69±0.12 |
| | 2 | 95.0 | 0.69±0.26 |
| | 3 | 95.0 | 1.20±0.20 |
| | 4 | 84.0 | 1.43±0.17 |
| | 5 | 89.0 | 2.12±0.32 |
| | 6 | 88.0 | 1.83±0.18 |
| | 7 | 91.0 | 1.22±0.16 |
| | 8 | 91.0 | 0.76±0.13 |
| | 9 | 85.0 | 1.27±0.14 |
| AntMaze | 1 | 92.0 | 7.95±2.87 |
| | 2 | 94.0 | 4.53±0.60 |
| | 3 | 81.0 | 9.95±2.81 |
| | 4 | 83.0 | 9.72±1.01 |
| | 5 | 62.0 | 32.42±5.89 |
| | 6 | 64.0 | 24.63±5.57 |
| | 7 | 82.0 | 15.80±4.78 |
| | 8 | 91.0 | 4.58±0.57 |
| | 9 | 60.0 | 8.98±0.94 |

**Analysis.** In the Cube environment, our method achieves consistently strong performance, with execution success rates exceeding **90%** on most task templates. The low planning time (typically below 2 seconds) indicates that the framework efficiently handles the dynamics of the 6-DoF manipulator. Performance degradation is observed only in more complex templates (Types 4 and 9), which involve long-horizon temporal dependencies and nested STL tasks. These cases require more extensive search and multiple calls to the trajectory generator, slightly increasing the planning time while reducing the success rate due to accumulated modeling uncertainty. Nevertheless, the overall high success rate demonstrates that our framework generalizes well to this manipulation-like domains.

In contrast, the AntMaze environment presents a substantially greater challenge due to its high-dimensional locomotion dynamics and strong coupling between joint configurations and global motion. Here, the average success rate remains around **80–90%** for moderate templates but drops to around **60%** for the most complex ones (Types 5, 6, and 9). A closer examination reveals that these harder tasks often involve stringent temporal nesting and "avoid" constraints, which are more sensitive to execution noise and controller-induced deviation. The higher planning times-especially for long-horizon templates (e.g., Type 5 and 6)-reflect the reduced efficiency of the trajectory generation module in scenarios requiring longer trajectories due to more complex dynamics. Nevertheless, our method is still able to produce feasible trajectories in most cases within an acceptable time frame.

Overall, these results confirm that the proposed framework scales effectively from smooth, fully actuated manipulation systems to complex locomotion environments. The modular combination of decomposition, allocation, and diffusion-based trajectory generation allows efficient reasoning over

STL objectives, maintaining both computational efficiency and high task success rates across diverse dynamical regimes.

## F.5 Details of Comparative Experiment with Optimization-based Method

To further evaluate the success rate of the progress allocation module in our algorithm, we compare it against a widely-used optimization-based algorithm [4] in a custom-built simulation environment. The baseline algorithm is employed as a sound and complete solution to accurately assess the feasibility of randomly generated test cases.

**Experimental Settings.** The experiment is conducted within a bounded $10 \times 10$ square 2D plane containing a circular obstacle. The underlying system dynamics are modeled using a double integrator. The agent starts from a randomly generated position and must complete the randomly generated STL tasks by reaching the target region within the specified time interval.

In this experiment, the baseline algorithm is implemented using the open-source library `stlpy` [2] and has full knowledge of the environmental information and system dynamics, while our algorithm only has access to the trajectory dataset.

To generate the trajectory dataset, we randomly sample start and end points in the environment and use the baseline algorithm to solve reach-avoid tasks. This process produces 200,000 collision-free trajectories that satisfy the system dynamics, which are then used to train the diffusion model and the Time Predictor.

To further enhance the trajectory generator's ability to produce trajectories of varying lengths, we train two diffusion models with different horizons: one dedicated to generating shorter trajectories and the other specialized for longer trajectories. The first model is trained on trajectory segments of length 16 for shorter trajectories, while the second model uses segments of length 32 to improve generalization to longer trajectories.

We generate 200 feasible STL tasks for each template, as described in Section F.2. The deterministic baseline algorithm is used to ensure the feasibility of these tasks. However, for templates 4 and 5, which involve multi-layer nesting of temporal operators, the baseline algorithm fails to find solutions within an acceptable time. In these cases, we still employ our algorithm's progress allocation module to verify feasibility.

**Evaluation Metrics.** In addition to the **Execution Success Rate (SR)** and **Total Planning Time (T0)** metrics described in Section 5, we introduce an additional evaluation metric:

- **Progress Allocation Success Rate (SR0):** The proportion of cases where the progress allocation module successfully identifies a sequence of waypoints. This metric specifically measures the reliability of the progress allocation module in our algorithm.

Table F.4: Result of Experiment in Custom-built Environment. SR0: Progress Allocation Success Rate; SR: Execution Success Rate;

| Type | SR0(%) | SR(%) | Total Planning Time(s)↓ | |
|---|---|---|---|---|
| | | | ours | baseline |
| 1 | 96.0 | 93.5 | 0.99±0.08 | 3.82±1.44 |
| 2 | 98.0 | 96.5 | 0.81±0.03 | 6.30±1.36 |
| 3 | 96.0 | 89.0 | 1.26±0.05 | 31.60±10.46 |
| 4 | - | 78.5 | 2.10±0.26 | Timeout |
| 5 | - | 83.0 | 2.57±0.10 | Timeout |
| 6 | 97.5 | 69.5 | 2.87±0.12 | 24.23±6.39 |
| 7 | 80.0 | 73.5 | 1.80±0.08 | 7.71±3.50 |
| 8 | 89.5 | 89.0 | 0.82±0.03 | 106.58±82.19 |
| 9 | 81.0 | 72.0 | 1.61±0.06 | 151.19±78.82 |

**Analysis.** The experimental results are summarized in Table F.4. Since the optimization-based baseline is used as an expert solver to certify the feasibility of randomly generated STL tasks, its execution success rate (**SR**) is naturally 100% and thus omitted from Table F.4. Our algorithm achieves

consistently high success rates across test cases generated from diverse task templates. Notably, the **Execution Success Rate (SR)** exceeds 69% in all scenarios, demonstrating the algorithm's strong generalization capability for STL tasks.

For all templates except 4 and 5, the **Progress Allocation Success Rate (SR0)** exceeds **80%**, indicating that the progress allocation module is generally reliable, albeit slightly conservative.

Finally, by comparing the **Total Planning Time**, our algorithm significantly outperforms the optimization-based baseline algorithm, highlighting the efficiency of the task decomposition and planning framework employed in our approach. Notably, for templates 4 and 5, which involve multi-layered nested STL tasks, the baseline algorithm fails to find feasible solutions within a reasonable time. In contrast, our algorithm demonstrates both high success rates and high efficiency, even in these complex scenarios.

### F.6   Analysis of the Predictor-Generator-Controller Framework

In our framework, the Time Predictor serves as a crucial component that provides heuristic guidance on system reachability during progress allocation. By estimating the time required for transitions between states, it allows the allocation process to consider not only the logical satisfaction of STL constraints but also the underlying dynamical feasibility between consecutive waypoints, thereby creating favorable conditions for subsequent trajectory generation. The Time Predictor operates in close collaboration with the diffusion-based trajectory generator. Specifically, the predictor estimates the expected travel time between two states, and the generator then produces a trajectory of the corresponding length. Although the true time-to-reach can vary considerably due to stochasticity and unmodeled dynamics, a generator trained on trajectory segments of diverse lengths demonstrates strong generalization. In practice, even a moderately accurate statistical estimate from the predictor typically suffices to produce a dynamically feasible trajectory. Moreover, the feedback control layer further compensates for residual prediction errors, ensuring consistent execution despite model approximation. To empirically validate the effectiveness of this collaboration, we conducted an additional experiment in both Maze2D and AntMaze environments.

**Experimental Settings.**   For each environment, we randomly sampled 1000 start-goal pairs (with goal region radii ranging from 3% to 6% of the arena size). For each case, the Time Predictor estimated the required trajectory length from the start position to the center of the goal region; the trajectory generator then produced a trajectory of that length, which was subsequently executed using a PD controller (in Maze2D) or an inverse dynamics model (in AntMaze) under the strict time-synchronous control protocol described in Section 3.5 (with $k = 1$ in Maze2D and $k = 2$ in AntMaze). All modules were employed exactly as implemented in our main framework, without additional tuning.

**Results and Analysis.**   The resulting execution success rates, summarized in Table F.5, demonstrate that even in the more challenging AntMaze setting, the current predictor-generator-controller pipeline achieves high overall reliability. These results underscore the effectiveness of modular integration among prediction, generation, and control within our framework. Given this modular design, each component can be further improved independently-for example, by incorporating more accurate time prediction models [51, 52, 58] or more expressive trajectory generators-to handle increasingly complex dynamical systems. We leave such extensions as promising directions for future work.

Table F.5: Execution Success Rate (%) of the Predictor-Generator-Controller Pipeline in Different Environments. Each result is computed over 1000 randomly sampled start-goal pairs.

| Environment | Umaze | Medium | Large | AntMaze |
|---|---|---|---|---|
| **Execution Success Rate (%)** | 93.9 | 89.0 | 83.7 | 84.8 |

### F.7   Implementation Details of Experiments

#### F.7.1   Calculation of the Robustness Value

In our experiments, we compute the robustness values of execution trajectories using the open-source library `stlpy` [2], which implements quantitative semantics for Signal Temporal Logic

(STL). However, in environments such as Maze2D, AntMaze, and Cube, transitions between states often require relatively long trajectories. As a result, the corresponding STL task intervals become lengthy, leading to a substantial computational burden when evaluating robustness directly on the full-resolution trajectories—often exceeding the available computational resources.

To mitigate this issue, we introduce a *temporal sampling factor* $\eta$, which defines the mapping between the time scale of the STL task and the resolution of the system trajectory used for evaluation. Specifically, one discrete time step in the STL task corresponds to $\eta$ time steps in the system trajectory. When computing robustness, we sample one state every $\eta$ steps to obtain a *down-sampled trajectory*, and then evaluate robustness on this sampled sequence. Importantly, the STL formula used for evaluation is also temporally rescaled, with all time intervals divided by $\eta$, so that the robustness value is computed with respect to a temporally consistent but shorter-horizon STL specification. This procedure effectively reduces computational overhead while preserving the temporal and logical structure of the original task.

It is worth emphasizing that the parameter $\eta$ is *conceptually distinct* from the parameter $k$ introduced in the control protocol (Section 3.5). While $k$ determines the number of low-level control updates executed per planning step during runtime—thus linking planning and control frequencies—$\eta$ only affects the post-hoc evaluation of robustness by defining how densely the executed trajectory is sampled for STL computation.

### F.7.2 Experimental Parameter Settings

Some of the parameters involved in the experiments are listed below, and their specific values are shown in Table F.6:

Table F.6: Parameters Used in the Experiments

| Env | $N_{\max}$ | $H$ | $N$ | $\gamma$ | $\eta$ | $k$ |
|---|---|---|---|---|---|---|
| Maze2D-Umaze | 1 | 128 | 64 | 0.8 | 8 | 1 |
| Maze2D-Medium | 1 | 256 | 256 | 0.9 | 8 | 1 |
| Maze2D-Large | 1 | 384 | 256 | 1.1 | 8 | 1 |
| AntMaze | 1 | 512 | 512 | 1.0 | 12 | 2 |
| Cube | 1 | 128 | 64 | 1.2 | 4 | 1 |
| Custom-Built | 1 | 16&32 | 64 | 1 | 1 | 1 |

- **Maximum Number of Attempts** ($N_{\max}$): The maximum number of attempts for new state sampling in Algorithm 2.

- **Horizon** ($H$): The planning horizon used during the training of the diffusion model.

- **Total Denoise Steps** ($N$): The total number of steps in the denoising process.

- **Scaling Factor** ($\gamma$): Applied to the predicted mean trajectory length, used to control the conservativeness of the Progress Allocation Module, as described in Section 3.3.

- **Sampling Factor** ($\eta$): Used when computing the robustness value of trajectories, as described in Section F.7.1.

- **Control Frequency** ($k$): The number of low-level control updates executed per planning step during runtime, as described in Section 3.5.

## G  Proofs

### G.1  Proof of Lemma 1

*Proof.* We prove the statement by structural induction on $\varphi$. Throughout, we assume discrete time $t \in \mathbb{Z}_{\geq 0}$ and use the shorthand $\mathbf{s}_0 \vDash \mathcal{R}(a, b, \mu) \Leftrightarrow \exists t \in [a, b] : \mathbf{x}_t \vDash \mu$ and $\mathbf{s}_0 \vDash \mathcal{I}(a, b, \mu) \Leftrightarrow \forall t \in [a, b] : \mathbf{x}_t \vDash \mu$ (cf. (3)-(4)).

**Base cases.**

*(i)* $\varphi = \mathrm{F}_{[a,b]}\mu$. The decomposition yields $\mathbb{P}_\varphi = \{\mathcal{R}(\lambda, \lambda, \mu)\}$, $\mathbb{T}_\varphi = \{\lambda \in [a,b]\}$, $\Lambda_\varphi = \{\lambda\}$. By the Boolean semantics of F (Eq. (C.4)),

$$\mathbf{s}_0 \vDash \mathrm{F}_{[a,b]}\mu \;\Leftrightarrow\; \exists \lambda \in [a,b] \text{ s.t. } \mathbf{x}_\lambda \vDash \mu \;\Leftrightarrow\; \exists \lambda \in [a,b] \text{ s.t. } \mathbf{s}_0 \vDash \mathcal{R}(\lambda, \lambda, \mu),$$

which is exactly the desired form with $\boldsymbol{\lambda} = [\lambda] \in \mathcal{F}_\varphi$.

*(ii)* $\varphi = \mathrm{G}_{[a,b]}\mu$. The decomposition yields $\mathbb{P}_\varphi = \{\mathcal{I}(a,b,\mu)\}$ and $\Lambda_\varphi = \mathbb{T}_\varphi = \varnothing$. By Eq. (C.5),

$$\mathbf{s}_0 \vDash \mathrm{G}_{[a,b]}\mu \;\Leftrightarrow\; \forall t \in [a,b] : \mathbf{x}_t \vDash \mu \;\Leftrightarrow\; \mathbf{s}_0 \vDash \mathcal{I}(a,b,\mu),$$

which matches the target equivalence (no time variables).

*(iii)* $\varphi = \mu_1 \mathrm{U}_{[a,b]}\mu_2$. The decomposition yields $\mathbb{P}_\varphi = \{\mathcal{I}(0,\lambda,\mu_1), \mathcal{R}(\lambda, \lambda, \mu_2)\}$, $\mathbb{T}_\varphi = \{\lambda \in [a,b]\}$, $\Lambda_\varphi = \{\lambda\}$. By Eq. (C.6),

$$\mathbf{s}_0 \vDash \mu_1 \mathrm{U}_{[a,b]}\mu_2 \Leftrightarrow \exists \lambda \in [a,b]: \ \mathbf{x}_\lambda \vDash \mu_2 \ \wedge \ \forall t \in [0,\lambda] : \mathbf{x}_t \vDash \mu_1$$
$$\Leftrightarrow \exists \lambda \in [a,b]: \ \mathbf{s}_0 \vDash \mathcal{R}(\lambda, \lambda, \mu_2) \ \wedge \ \mathbf{s}_0 \vDash \mathcal{I}(0,\lambda,\mu_1),$$

again yielding the target form with $\boldsymbol{\lambda} = [\lambda] \in \mathcal{F}_\varphi$.

**Inductive steps.** For each constructor below we assume the lemma holds for the immediate subformula(e).

*(1) Conjunction* $\varphi = \varphi_1 \wedge \varphi_2$. By semantics, $\mathbf{s}_0 \vDash \varphi \Leftrightarrow (\mathbf{s}_0 \vDash \varphi_1 \ \wedge \ \mathbf{s}_0 \vDash \varphi_2)$. The decomposition uses disjoint unions $\mathbb{P}_\varphi = \mathbb{P}_{\varphi_1} \uplus \mathbb{P}_{\varphi_2}$, $\mathbb{T}_\varphi = \mathbb{T}_{\varphi_1} \uplus \mathbb{T}_{\varphi_2}$, $\Lambda_\varphi = \Lambda_{\varphi_1} \uplus \Lambda_{\varphi_2}$, implicitly $\alpha$-renaming to avoid clashes. Since each constraint in $\mathbb{T}_{\varphi_i}$ involves a single variable, the feasible set factors: $\mathcal{F}_\varphi = \mathcal{F}_{\varphi_1} \times \mathcal{F}_{\varphi_2}$. By the IH on $\varphi_1, \varphi_2$ and the fact that every progress in $\mathbb{P}_\varphi$ depends only on its own component of $\boldsymbol{\lambda} = [\boldsymbol{\lambda}_1, \boldsymbol{\lambda}_2]$, we obtain

$$\mathbf{s}_0 \vDash \varphi \;\Leftrightarrow\; \exists \boldsymbol{\lambda} \in \mathcal{F}_\varphi \text{ s.t. } \forall \mathcal{P} \in \mathbb{P}_\varphi : \ \mathbf{s}_0 \vDash \mathcal{P}(\boldsymbol{\lambda}).$$

*(2) Outer finally* $\varphi' = \mathrm{F}_{[a,b]}\varphi$. By Eq. (C.4), $\mathbf{s}_0 \vDash \varphi' \Leftrightarrow \exists \lambda \in [a,b] : \mathbf{s}_\lambda \vDash \varphi$. **Time-shift identity.** For any progress $\mathcal{P}(c,d,\mu)$ and any $k \in \mathbb{Z}_{\geq 0}$,

$$\mathbf{s}_k \vDash \mathcal{P}(c,d,\mu) \;\Leftrightarrow\; \mathbf{s}_0 \vDash \mathcal{P}(c+k, d+k, \mu). \tag{G.1}$$

This is immediate from the definitions of $\mathcal{R}$ and $\mathcal{I}$. The decomposition introduces a fresh $\lambda \in [a,b]$ and shifts every progress of $\varphi$ by $+\lambda$; hence $\Lambda_{\varphi'} = \Lambda_\varphi \uplus \{\lambda\}$ and $\mathcal{F}_{\varphi'} = \{[\boldsymbol{\lambda}, \lambda] \mid \boldsymbol{\lambda} \in \mathcal{F}_\varphi, \ \lambda \in [a,b]\}$. Applying the IH to $\varphi$ on the shifted signal $\mathbf{s}_\lambda$ and then using (G.1) yields

$$\mathbf{s}_0 \vDash \varphi' \;\Leftrightarrow\; \exists \boldsymbol{\lambda}' \in \mathcal{F}_{\varphi'} \text{ s.t. } \forall \mathcal{P}' \in \mathbb{P}_{\varphi'} : \ \mathbf{s}_0 \vDash \mathcal{P}'(\boldsymbol{\lambda}').$$

(The special case $\mathrm{F}_{[a,a]}$ is the same with a constant shift $+a$ and no new variable.)

*(3) Outer always* $\varphi' = \mathrm{G}_{[a,b]}\varphi$. In discrete time,

$$\mathrm{G}_{[a,b]}\varphi \;\equiv\; \bigwedge_{k=a}^{b} \mathrm{F}_{[k,k]}\varphi, \tag{G.2}$$

since $\mathbf{s}_0 \vDash \mathrm{G}_{[a,b]}\varphi$ iff $\mathbf{s}_k \vDash \varphi$ for every integer $k \in [a,b]$. The decomposition makes an independent copy $(\mathbb{P}_\varphi^{(k)}, \mathbb{T}_\varphi^{(k)}, \Lambda_\varphi^{(k)})$ for each $k$ and shifts every progress by $+k$, then merges all copies:

$$\mathbb{P}_{\varphi'} = \bigcup_{k=a}^{b}\{\mathcal{P}(c_\Lambda + k, d_\Lambda + k, \mu) \mid \mathcal{P}(c_\Lambda, d_\Lambda, \mu) \in \mathbb{P}_\varphi^{(k)}\}, \quad \Lambda_{\varphi'} = \biguplus_{k=a}^{b} \Lambda_\varphi^{(k)}, \quad \mathcal{F}_{\varphi'} = \prod_{k=a}^{b} \mathcal{F}_\varphi.$$

Applying item (2) to each $\mathrm{F}_{[k,k]}$ factor and using (G.1) yields the target form. *Merging constant invariances.* If $\mathcal{I}(c,d,\mu) \in \mathbb{P}_\varphi$ has constant endpoints, then its shifted copies $\{\mathcal{I}(c+k, d+k, \mu)\}_{k=a}^{b}$ are equivalent to the single $\mathcal{I}(c+a, d+b, \mu)$ because $\bigcup_{k=a}^{b}[c+k, d+k] = [c+a, d+b]$ in discrete time; hence merging preserves truth.

*(4) Outer until* $\varphi' = \phi \mathrm{U}_{[a,b]}\varphi$. By assumption, $\phi$ contains no F or U, thus its decomposition consists solely of invariance progresses with constant endpoints and $\Lambda_\phi = \mathbb{T}_\phi = \varnothing$. The decomposition

of $\varphi'$: introduce $\lambda \in [a, b]$, shift each progress of $\varphi$ by $+\lambda$, and prolong every $\mathcal{I}(c, d, \mu) \in \mathbb{P}_\phi$ to $\mathcal{I}(c, d + \lambda, \mu)$. Formally,

$$\mathbb{P}_{\varphi'} = \{\mathcal{P}(c_\Lambda + \lambda, d_\Lambda + \lambda, \mu) \mid \mathcal{P} \in \mathbb{P}_\varphi\} \cup \{\mathcal{I}(c, d + \lambda, \mu) \mid \mathcal{I}(c, d, \mu) \in \mathbb{P}_\phi\},$$
$$\mathcal{F}_{\varphi'} = \{[\boldsymbol{\lambda}, \lambda] \mid \boldsymbol{\lambda} \in \mathcal{F}_\varphi, \ \lambda \in [a, b]\}.$$

By Eq. (C.6),

$$\mathbf{s}_0 \vDash \phi \mathrm{U}_{[a,b]} \varphi \ \Leftrightarrow \ \exists \lambda \in [a, b] \text{ s.t. } \Big(\mathbf{s}_\lambda \vDash \varphi \ \wedge \ \forall \tau \in [0, \lambda] : \ \mathbf{s}_\tau \vDash \phi\Big). \tag{G.3}$$

The first conjunct is handled by item (2) with the shift identity (G.1). For the second conjunct, note the following **sliding-invariance identity**: for any constant $c \leq d$,

$$\big(\forall \tau \in [0, \lambda] : \ \mathbf{s}_\tau \vDash \mathcal{I}(c, d, \mu)\big) \ \Leftrightarrow \ \mathbf{s}_0 \vDash \mathcal{I}(c, d + \lambda, \mu), \tag{G.4}$$

because $\mathbf{s}_\tau \vDash \mathcal{I}(c, d, \mu)$ means $\forall t \in [c + \tau, d + \tau] : \mathbf{x}_t \vDash \mu$, and $\bigcup_{\tau=0}^\lambda [c + \tau, d + \tau] = [c, d + \lambda]$ in discrete time. Since $\mathbb{P}_\phi$ is a conjunction of such invariances, applying (G.4) to each member yields the prolonged invariances in $\mathbb{P}_{\varphi'}$. Combining these observations with (G.3) gives

$$\mathbf{s}_0 \vDash \varphi' \ \Leftrightarrow \ \exists \boldsymbol{\lambda}' \in \mathcal{F}_{\varphi'} \text{ s.t. } \forall \mathcal{P}' \in \mathbb{P}_{\varphi'} : \ \mathbf{s}_0 \vDash \mathcal{P}'(\boldsymbol{\lambda}').$$

**Conclusion.** All constructors preserve the claimed equivalence. By structural induction, for any PNF STL formula without disjunctions,

$$\mathbf{s}_0 \vDash \varphi \ \Longleftrightarrow \ \exists \boldsymbol{\lambda} \in \mathcal{F}_\varphi \text{ s.t. } \forall \mathcal{P} \in \mathbb{P}_\varphi : \ \mathbf{s}_0 \vDash \mathcal{P}(\boldsymbol{\lambda}).$$

If $\Lambda_\varphi = \varnothing$, then $\mathcal{F}_\varphi = \{\emptyset\}$ and the statement reduces to $\mathbf{s}_0 \vDash \varphi \Longleftrightarrow \forall \mathcal{P} \in \mathbb{P}_\varphi : \ \mathbf{s}_0 \vDash \mathcal{P}$. $\qquad \square$

## G.2 Proof of Lemma 2

*Proof.* We argue by maintaining inductive invariants along the DFS. At each node of the search tree we keep a tuple $(\mathbf{x}, t, \mathbb{P}_{\mathrm{rem}}^\mathcal{R}, \mathbb{T}, \tilde{\mathbf{s}})$, where $\tilde{\mathbf{s}}$ is the waypoint list accumulated so far and $\mathbb{P}_{\mathrm{rem}}^\mathcal{R}$ are the remaining reachability progresses.

**Inductive invariants.** After initialization and after every successful extension by Algorithm 2 and UpdateConstraint, the following hold:

(I1) *Feasibility is preserved:* the current constraint set $\mathbb{T}$ admits at least one assignment of time variables (i.e., the feasible set is nonempty).

(I2) *Committed reachabilities are honored:* for every reachability progress $\mathcal{R}(a_\Lambda, b_\Lambda, \mu)$ that has been assigned a waypoint $(\mathbf{x}', t')$ and removed from $\mathbb{P}_{\mathrm{rem}}^\mathcal{R}$, $\mathbb{T}$ contains the inequalities $a_\Lambda \leq t'$ and $b_\Lambda \geq t'$. Hence any $\boldsymbol{\lambda}$ feasible for $\mathbb{T}$ satisfies $t' \in [a_\Lambda(\boldsymbol{\lambda}), b_\Lambda(\boldsymbol{\lambda})]$.

(I3) *Waypoint-level invariance consistency:* let $\mathcal{I}(c, d_\Lambda, \mu')$ be any invariance progress whose *start time has been determined* (by preprocessing every $\mathcal{I}$ has a preceding $\mathcal{R}(c, c, \mu')$; once that $\mathcal{R}$ is assigned at time $c = t^*$, the start is fixed). Whenever a new waypoint $(\mathbf{x}', t')$ is appended with $\mathbf{x}' \nvDash \mu'$, Algorithm 2 computes a conflict interval $\mathcal{O}$ and UpdateConstraint adds $d_\Lambda < t'$; consequently, no feasible $\boldsymbol{\lambda}$ can make the interval $[c, d_\Lambda(\boldsymbol{\lambda})]$ cover $t'$, so $\mathcal{I}$ cannot be active at $t'$.

**Initialization.** At the start, $\tilde{\mathbf{s}} = [(\mathbf{x}_0, 0)]$ and $\mathbb{T}$ is the given constraint set; by assumption it is feasible, so (I1) holds. No progress has been assigned, so (I2) and (I3) are vacuous.

**Preservation under one extension.** Suppose the algorithm chooses a remaining reachability $\mathcal{R}(a_\Lambda, b_\Lambda, \mu)$ at $(\mathbf{x}, t)$ and calls Algorithm 2. By construction, the routine computes the largest currently feasible time window $[t_{\mathrm{min}}, t_{\mathrm{max}}]$ consistent with $\mathbb{T}$, and samples a candidate $\mathbf{x}' \models \mu$ together with a predicted arrival time $t'$. If $t' > t_{\mathrm{max}}$ or the residual feasible window is covered by conflicts, the attempt is discarded; otherwise the routine returns the *earliest* nonconflicting time $t_{\mathrm{new}} \in [\max\{t', t_{\mathrm{min}}\}, t_{\mathrm{max}}] \setminus \mathcal{O}$. Therefore, with $t^* := t_{\mathrm{new}}$, the augmented constraints $a_\Lambda \leq t^*$, $b_\Lambda \geq t^*$ are consistent with the current $\mathbb{T}$, establishing (I1) and (I2) at the child node. Moreover, for every invariance with determined start time that is violated by $\mathbf{x}'$, UpdateConstraint inserts $d_\Lambda < t^*$; since Algorithm 2 avoided $\mathcal{O}$ and $t^*$ lies strictly to the right of the current lower bound for $d_\Lambda$, the new inequality is compatible with $\mathbb{T}$. Hence (I3) holds as well.

**Termination case and witness construction.** When the algorithm returns, $\mathbb{P}^{\mathcal{R}}_{\text{rem}} = \varnothing$ and the final constraints are $\mathbb{T}_f$ with feasible set $\mathcal{F}_f \neq \varnothing$ by (I1). Pick any $\boldsymbol{\lambda} \in \mathcal{F}_f$. By (I2), for every assigned reachability with waypoint time $t_i$, we have $t_i \in [a_\Lambda(\boldsymbol{\lambda}), b_\Lambda(\boldsymbol{\lambda})]$, hence $\mathbf{s}_0 \models \mathcal{R}(a_\Lambda(\boldsymbol{\lambda}), b_\Lambda(\boldsymbol{\lambda}), \mu)$. For any invariance $\mathcal{I}(a_\Lambda, b_\Lambda, \mu')$ and any waypoint time $t_i$ that lies in its active window $[a_\Lambda(\boldsymbol{\lambda}), b_\Lambda(\boldsymbol{\lambda})]$, there are two possibilities: (i) $\tilde{\mathbf{x}}_i \models \mu'$, in which case the waypoint trivially satisfies the invariance; or (ii) $\tilde{\mathbf{x}}_i \not\models \mu'$, in which case the start of this invariance has already been determined by its preceding reachability, and (I3) guarantees that $t_i$ cannot lie in the interval (because $d_\Lambda < t_i$ was added). Either way, the stated waypoint-level invariance condition holds. This proves the lemma. □

**Remark 3** (Scope of the guarantee). *The lemma certifies that, under some feasible time-variable assignment, all* reachability *progresses are satisfied and no* invariance *progress is violated at the returned waypoint times. It does not claim invariance satisfaction at* every *intermediate time step between waypoints.*

### G.3 Proof of Theorem 1

*Proof.* Let $(\mathbb{P}_\varphi, \mathbb{T}_\varphi)$ be the progress/constraint pair returned by the Semantics-based Decomposition, and let $\tilde{\mathbf{s}} = (\tilde{\mathbf{x}}_0, t_0)(\tilde{\mathbf{x}}_1, t_1) \ldots (\tilde{\mathbf{x}}_n, t_n)$ with $0 = t_0 \leq \cdots \leq t_n$ and final constraint set $\mathbb{T}_f$ be the output of the Progress Allocation algorithm. By assumption, $\mathcal{F}_f \neq \varnothing$. Fix any feasible assignment $\boldsymbol{\lambda}^\star \in \mathcal{F}_f$ and let $\mathbf{s}_0 = \mathbf{x}_0 \mathbf{x}_1 \ldots \mathbf{x}_T$ ($T \geq t_n$) be the signal induced by the trajectory $\boldsymbol{\tau}$ returned by the Trajectory Generation module, which (by assumption) satisfies: (i) $\mathbf{x}_{t_i} = \tilde{\mathbf{x}}_i$ for all $i$, and (ii) for every invariance $\mathcal{I}(a_\Lambda, b_\Lambda, \mu)$, $\forall t \in [a_\Lambda(\boldsymbol{\lambda}^\star), b_\Lambda(\boldsymbol{\lambda}^\star)]$ we have $\mathbf{x}_t \models \mu$.

**Reachability progresses.** By construction of the allocation phase and by Lemma 2, for every $\mathcal{R}(a_\Lambda, b_\Lambda, \mu) \in \mathbb{P}_\varphi$ there exists an index $i$ such that (a) $\tilde{\mathbf{x}}_i \models \mu$ and (b) $t_i \in [a_\Lambda(\boldsymbol{\lambda}^\star), b_\Lambda(\boldsymbol{\lambda}^\star)]$, because $\mathbb{T}_f$ contains the constraints $a_\Lambda \leq t_i$ and $b_\Lambda \geq t_i$, which are satisfied by $\boldsymbol{\lambda}^\star$. Since the generated signal visits all waypoints, $\mathbf{x}_{t_i} = \tilde{\mathbf{x}}_i$, hence $\mathbf{s}_0 \models \mathcal{R}(a_\Lambda(\boldsymbol{\lambda}^\star), b_\Lambda(\boldsymbol{\lambda}^\star), \mu)$.

**Invariance progresses.** By property (ii) of the generated trajectory, for every $\mathcal{I}(a_\Lambda, b_\Lambda, \mu) \in \mathbb{P}_\varphi$ we have $\forall t \in [a_\Lambda(\boldsymbol{\lambda}^\star), b_\Lambda(\boldsymbol{\lambda}^\star)]$: $\mathbf{x}_t \models \mu$, i.e., $\mathbf{s}_0 \models \mathcal{I}(a_\Lambda(\boldsymbol{\lambda}^\star), b_\Lambda(\boldsymbol{\lambda}^\star), \mu)$.

**Conclusion via decomposition soundness.** We have shown that for the same feasible assignment $\boldsymbol{\lambda}^\star$,
$$\forall \mathcal{P} \in \mathbb{P}_\varphi: \quad \mathbf{s}_0 \models \mathcal{P}(\boldsymbol{\lambda}^\star).$$
Therefore, by Lemma 1, $\mathbf{s}_0 \vDash \varphi$. This proves the theorem. □

