# OpenReview forum: "Zero-Shot Trajectory Planning for Signal Temporal Logic Tasks"
_NeurIPS.cc/2025/Conference — NeurIPS 2025 poster_

### Official Review · Reviewer_fjcC · 2025-06-24

**Clarity:** 2
**Significance:** 4
**Originality:** 3
**Rating:** 5
**Confidence:** 4

**Summary:**

This paper presents a novel, data-driven hierarchical framework for generating robot trajectories that satisfy complex Signal Temporal Logic (STL) tasks when the system dynamics are unknown. The proposed method decomposes the STL specification into simpler timed objectives, searches for waypoints to achieve them, and then uses a pre-trained diffusion model to generate and stitch together the final trajectory segments. This approach allows for zero-shot generalization to new tasks using only task-agnostic data and is formally proven to guarantee the satisfaction of the given STL specification if a solution is generated. Run-time execution performance is shown to be quicker than existing diffusion-based solutions with a higher overall success rate.

**Questions:**

1. In Algorithm 2, L7, what is $d^{\text{min}}_{det}$? On a related note, I find it difficult to understand the purpose of L5-9 in Algorithm 2, could I get an explanation please or a reference to where it is explained in detail?
2. Would it be possible to include experiments with more complicated dynamics such as the Ant robot?
3. In the Results section (L411-412), why is the performance of the method underplayed? It appears to have a greater than 90% and 79% overall success rate (vs 80 and 69 as mentioned).
4. In Table G.1, why is the success rate of STLPY not mentioned?
5. How does the approach manage discrepancies between planning times and execution delays (i.e., failure to reach a goal on time due to the controller)? How does this impact the success rate?
6. Typos:
    - L202: way-point → way-points
    - L207: satisfication → satisfaction
    - L252: fucntion → function
    - L681: Sementic → Semantic

**Ethical Concerns:**

["NO or VERY MINOR ethics concerns only"]

**Final Justification:**

The additional post-rebuttal study on the `TimePredict` function as a measure of the system dynamics helped alleviate some concerns of its ability. The approach is still largely heuristic-based and an option to modify the optimality criteria would have been ideal. However, as it is, the paper provides an interesting contribution to field of STL-guided planning. A standardized open-source implementation for future work to compare against would be a useful contribution.

**Limitations:**

- Assumes a valid $\texttt{TimePredict}$ function exists for the entire system, but this is not easy in many environments. For example, a setting where multiple robots are placed in a maze, predicting the time between global system states could be hard due to conflicts.
- The algorithm heuristics may lead to sub-optimal solutions in terms of makespan which is not fully examined.
- Delays during execution can cause the specification to not be satisfied and this does not appear formalized.

**Quality:**

3

**Strengths And Weaknesses:**

## Strengths:

- **Novelty:** The algorithm, while being manually constructed and heuristic based, focuses on generating a valid solution to a given STL specification using Diffusion-based sampling. The approach is novel and cleverly avoids the problem of getting long range trajectories with diffusion models (by sampling between generated waypoints) as well as handling Invariance or Avoidance Predicates (by using $\texttt{SafeDiffuser}$ that integrates Control Barrier Functions while sampling).
- **Speed:** The algorithm is shown to be much faster than existing approaches and elegantly handles the slow inference time of diffusion models.

## Weaknesses:

- **Assumption of TimePredict:** The accuracy of $\texttt{TimePredict}$ is important and is hard to guarantee in a general setting. For example, depending on the maze structure, the time between states may vary drastically. Some requirements on this function (such as assumed accuracy range) along with related experiments (in a given maze) would be helpful to understand the validity of the approach and how easy it would be to get this function. This is especially the case in environments with complex dynamics (e.g., instead of a point mass as in the experiments, if the agent was a quadruped or a differential drive robot).
- **Solution Optimality:** The approach has several heuristics and assumptions that guide the solution generation yet may not be optimal in many scenarios. Consider Algorithm 1 L9 For each progress ordered by starting time $a_\Lambda$ - this may cause poor solutions in terms of makespan, e.g. $\phi = F_{[0,100]} {A}\land F_{[1,100]}{B}$ where goal A is far away and goal B is near the starting state. This would yield a solution visiting the far goal A first and the coming back for goal B. I realize there are no mentions of solution optimality by the given algorithm, but this is just pointing out the fact that different solutions have distinct characteristics which are often of concern to the user.
- **Metrics Beyond Success Rate:** On a related note, rather than success rate alone, a makespan equivalent metric or “time to complete” metric could help to give a better idea of solution quality rather than simply success rate.
- **Writing:** Although the ideas proposed are interesting, the writing and organization could do with several refinements.
    - **Organization of Section 3:** At over four pages, Section 3 is overly long, making it difficult to follow. Subsection 3.3, in particular, should be reorganized into distinct subsections for each primary module of the algorithm (e.g., Main-Allocation, $\texttt{SampleState}$) to clearly delineate their functions.
    - **Clarity of Notation:** I found the notation in L163-L174 somewhat unclear. While going through the appendix provided the necessary clarification, I suggest incorporating some of those details into the main body to improve readability.  For example, the definition of $\Lambda$ (L163) being a set of time instance variables (clarified in L236) and the domain of $\mathfrak{t}\in\mathbb{T}_\varphi$ (L171) could be made more explicit in the main text.
    - **Correction of References:** L 283 mentions “the above DFS” which erroneously points to Algorithm 2, while I believe this references Algorithm 1 (a reference could be inserted here). Additionally, this line could be in the suggested new Main-Allocation subsection which can be separate from the $\texttt{SampleState}$ subsection.
    - **Refinement of Figure 1:** The figure lacks clarity and consistency with the text. Labels in the diagram, such as "Guided Sampling," should be updated to match section titles like "Trajectory Generation." The diagram itself should also employ standard conventions, using different shapes to distinguish processes from outputs (e.g., "Progress Allocation" vs. "Waypoints Sequence").

---

> ### Author Rebuttal · Authors · 2025-07-31
>
> We scincerely thank you for your very constructive feedback. In particular, your suggestions to analyse the feasibility of the TimePredictor, include additional metrics, and conduct experiments in more challenging environments guided us to broaden and strengthen our evaluation. We also appreciate your acknowledgement of the novelty of our approach.
>
> **R4.1 TimePredict accuracy is critical and might be unreliable in complex domains.**
>
> In our framework, the TimePredictor collaborates with the trajectory generator: the TimePredictor estimates the realistic travel time between states, and the trajectory generator produces a trajectory of that length. Admittedly, time‑to‑reach can vary drastically. However, a generator trained on trajectories of diverse lengths exhibits sufficient generalization; in practice, even a moderately accurate statistical estimate from the Time Predictor is usually adequate for producing a feasible trajectory.
>
> To illustrate this collaboration, we conducted a simple test in **Maze2d** and **antmaze** (will be introduced in R4.6). For each environment we created **1 000** random start‑and‑goal cases (goal region radius = 3 %–6 % of the arena size). For each case, we used TimePredictor to estimate the required trajectory length from the start to the center of the goal region, then generated a trajectory of that length using the trajectory generator, and finally tracked it with a PD controller (in Maze2d) or an inverse dynamics model (in antmaze). All modules were used as implemented in our method. The execution success rates shown in the following Table indicate that even in the challenging antmaze setting, our current modules achieve high success. Given our modular design, each component can be upgraded independently for more complex scenarios. We will further dissect module contributions and explore potential improvements in future work.
> | Maze2d-Umaze|Maze2d-Medium|Maze2d-Large|Antmaze-Medium|
> |-|-|-|-|
> |93.9%|89.0%|83.7%|84.8%|
>
> **R4.2 Lack of optimality**
>
> Note that our heuristic order (lines 283–285) is as follows: we prioritize progresses with earlier potential **deadlines**. If multiple candidates share the same deadline, we prefer the one with the earlier potential start time. Therefore, in this example, the algorithm will attempt to complete goal **B** first, because its deadline is earlier.
>
> Our current objective is **task correctness plus dynamic feasibility**, which is already challenging under unknown dynamics. We agree that incorporating additional metrics (e.g., makespan) is a promising research direction. One feasible approach to achieve this may involve replacing our current “stop at the first solution” mode with an anytime mode, allowing it to obtain better solutions by increasing the search effort. We appreciate this suggestion and will add a related discussion in the revised version, as well as investigate this direction in future work.
>
> **R4.3 Evaluation metrics: success rate alone is insufficient.**
>
> As explained above, we do not currently focus on additional optimization objectives. However, we agree that introducing quantitative metrics beyond success rate to assess solution quality is a valuable suggestion. Besides success rate, we now report the **average STL robustness value** of executed trajectories (top & bottom 5 % trimmed) in Maze2d. Our method consistently outperforms RGD. We will include these results and the corresponding analysis in the revised manuscript.
>
> |Env|Template|RGD|Ours|
> |-|-|-|-|
> |U | 1 | 0.1084 ± 0.1132 |0.1938 ± 0.0715|
> |U | 2 | -0.2208 ± 0.2826|0.1504 ± 0.0814|
> |U | 3 | -0.1695 ± 0.2521|0.1354 ± 0.0745|
> |M | 1 | 0.0885 ± 0.2768 |0.3205 ± 0.0957|
> |M | 2 | -0.2277 ± 0.3089|0.2013 ± 0.1950|
> |M | 3 | -0.2393 ± 0.3130|0.1761 ± 0.2060|
> |L | 1 | -0.1927 ± 0.3198|0.3180 ± 0.1228|
> |L | 2 | -0.3926 ± 0.2284|0.1672 ± 0.2411|
> |L | 3 | -0.2886 ± 0.2934|0.1639 ± 0.2286|
>
> **R4.4 Writing & presentation.**
>
> Thank you for the suggestions. We will streamline Section 3, clarify the notation, fix the reference and figure labels, and correct all typos to improve readability in the revision.
>
> **R4.5 About Algorithm 2.**
>
> In Algorithm 2, line 7, $d_{\Lambda,\mathbb{T}}^{min}$ denotes the earlist possible end time of the invariance progress $\mathcal{I}(c,d_\Lambda,\mu)$ under time‐variable constraints $\mathbb{T}$. Analogous to $a_{\Lambda,\mathbb{T}}^{min}$, it can be obtained by solving an ILP as described in lines 230–235.
>
> For lines 5–9 of Algorithm 2, we first correct a typo by renaming the $\mu$ in $\mathcal{I}(c,d_\Lambda,\mu)$ to $\mu'$ to distinguish the predicates. The purpose of this segment is to accumulate time intervals during which the state $x'$ is forbidden. Concretely, for each active invariance progress $\mathcal{I}(c,d_\Lambda,\mu')$, we check whether $x'$ satisfies $\mu'$. If not, it conflicts with that invariance progress, implying that $x'$ if forbidden while the invariance progress is active. We then compute the earliest possible end time $d_{\Lambda,\mathbb{T}}^{min}$ for that progress and add the interval $[c,\;d_{\Lambda,\mathbb{T}}^{min}]$ to the conflict intervals.
>
> **R4.6 Experiments with more complicated dynamics**
>
> We have **extended our evaluation to two dynamics‑rich tasks** from the offline goal‑conditioned RL benchmark **OGBench** [1]: *cube-single-play* (“cube”) and *antmaze-medium-navigate* (“antmaze”). The cube environment involves controlling a 6-DoF UR5e robot arm to manipulate cubes, whereas the antmaze environment involves controlling an 8-DoF Ant agent to navigate in a maze.
>
> In our experiments, we train exclusively on the STL task-agnostic trajectory datasets provided by OGBench, but replace the original goal-conditioned tasks with STL tasks. Specifically, in the antmaze environment, we adopt the same task formulation as in Sec. 5, requiring the agent to reach specified regions to satisfy the STL formula. In the cube environment, we focus solely on the end-effector position, requiring it to visit designated regions to satisfy the STL task (ignoring cube manipulation). In the cube environment, following the settings in our experiment, we first generate trajectories in the $x$–$y$–$z$ space and then track them with a PD controller. In the antmaze environment, following [2], we plan trajectories in the $x$–$y$ space and track with an inverse dynamics model that takes a 29-dimensional observation and the next $x$–$y$ goal as input and outputs an 8-dimensional action at each step.
>
> As in Sec. 5, we generate 100 test cases for each of the nine task templates (Table F.1) in both environments and report the execution success rates in the following Table. These results (over 60 % in antmaze and over 85 % in cube) confirm that our framework remains effective under substantially more complex dynamics.
>
> A more detailed description of the experimental setup, along with comprehensive results and analysis, will be provided in the revised manuscript.
>
> |Environment|Template|Execution Success Rate (%)|
> |-|-|-|
> |antmaze|1|**91.0**|
> |antmaze|2|**97.0**|
> |antmaze|3|**84.0**|
> |antmaze|4|**63.0**|
> |antmaze|5|**71.0**|
> |antmaze|6|**63.0**|
> |antmaze|7|**81.0**|
> |antmaze|8|**87.0**|
> |antmaze|9|**67.0**|
> |cube|1|**98.0**|
> |cube|2|**100.0**|
> |cube|3|**98.0**|
> |cube|4|**98.5**|
> |cube|5|**85.0**|
> |cube|6|**90.0**|
> |cube|7|**97.0**|
> |cube|8|**100.0**|
> |cube|9|**99.0**|
>
> Our current implementation adopts **simple baseline modules** for time prediction, trajectory generation, and low-level control. In highly challenging scenarios such as **antmaze**, these basic choices become potential performance bottlenecks. Since our framework is designed to be modular, each component can be replaced by a stronger alternative. Recent advances in diffusion-based trajectory planning (e.g., [2, 3]) provide promising drop-in candidates. We will integrate these enhanced modules and extend the evaluation to even more demanding environments to quantify their benefit in future work.
>
> **R4.7 Why is the performance of the method underplayed**
>
> Table 2 shows only a subset of results due to page limits; full statistics are in Table F.2. Values quoted in the text are aggregate summaries of the complete table. We will clarify this in the revised version.
>
> **R4.8 Why is the success rate of STLPY not mentioned?**
>
> STLPY is a *sound and complete* planner with full dynamics and map access, so every randomly generated test case is conditioned to be solvable by STLPY; its success rate is therefore **100 %**. We omitted it for brevity but will add this explanation.
>
> **R4.9 Discrepancies between planning times and execution delays**
>
> Our system outputs one control action per planned step. If tracking error prevents reaching a waypoint in time, the STL specification may be violated. The execution success rates in our experiments indeed reflects this kind of failure modes.
>
> These kind of execution failures can arise from:
> - **Controller limits** (e.g., a sub-optimal inverse-dynamics model).
> - **Insufficient dynamic feasibility** of the planned trajectory.
>
> Potential future remedies include:
>
> - Improving the TimePredictor and diffusion-based trajectory generator to produce more dynamically feasible plans.
> - Introducing **online re-planning** when the deviation exceeds a predefined threshold (Sec. F.1).
> - Allowing low-level controllers to execute multiple sub-steps per planned state, as long as the total time remains within the discrete-time budget.
>
> We will elaborate on these points in the revised version.
>
> > [1] Park S, Frans K, Eysenbach B and Levine S (2025) Ogbench: Benchmarking offline goal-conditioned rl. In: International Conference on Learning Representations.
> >
> > [2] Luo Y, Mishra U A, Du Y, et al. Generative trajectory stitching through diffusion composition[J]. arXiv:2503.05153, 2025.
> >
> > [3] Chen C, Hamed H, Baek D, et al. Extendable long-horizon planning via hierarchical multiscale diffusion[J].arXiv:2503.20102, 2025.

---

> > ### Comment · Reviewer_fjcC · 2025-08-06
> >
> > I appreciate the detailed response by the authors’ and the additional examination of `TimePredict` and its efficiency.
> >
> > Your new tables prompted a clarifying question regarding the path-following implementation. If the PD controller was used to follow the generated path, was the target waypoint updated only after the previous one was reached, or was a time-based shift used? This points to the question about the planning and execution gaps when following a generated plan that is not explicitly verified with the system's dynamics.
> >
> > In Table G.1, it would also be helpful to mention the specific controller used to evaluate the execution success rate.
> >
> > Note: I have edited the times in my original example to better clarify my point about solution optimality.

---

> > > ### Author Response · Authors · 2025-08-07
> > >
> > > Thank you for continuing the discussion and for the valuable follow-up questions.
> > >
> > >
> > > ### About Path-following implementation
> > >
> > >   Our experiments treat the system as discrete-time. The *control* step and the *planning* step share the **same period**, so one control action is issued for every planned state.
> > >
> > >   At step $i$ the controller aims for the next planned state $x_i$ **regardless of whether $x_{i-1}$ was exactly reached**. Execution therefore advances strictly by time, not by waypoint proximity. This guarantees that the realised trajectory respects the STL time windows.
> > >   **Example:** For $F_{[0,10]} A \wedge F_{[0,10]} B$ with timed waypoints $(x_0,0)\to(x_a,3)\to(x_b,8)$, the generator yields $\tau=x_0x_1\ldots x_8$ (length 9). During deployment we execute **exactly 8** control steps; at each step $i$ the controller references $x_i$. Even if $x_a$ is missed at step 3, the system proceeds to track $x_4$ at step 4, etc.
> > >
> > >   With this strict schedule, both the PD controller (Maze2d) and the inverse-kinematics controller (Antmaze) achieve high tracking success, indicating that the planned trajectories are dynamically feasible.
> > >
> > >   In real deployments where the low-level control cycle is faster than the planning step, the controller can take several micro-steps toward the next high-level target, provided the cumulative time still fits the discrete plan.
> > >
> > >   We will add this clarification to the paper and specify the exact controller (PD or inverse-kinematics) in Table G.1 and related tables.
> > >
> > >
> > > ### On solution optimality
> > >
> > > We fully endorse this revised example. We reaffirm that our current goal is **task correctness plus dynamic feasibility**; makespan or other optimality criteria remain open. An attractive direction is an **anytime allocator** that continues searching after the first feasible plan to trade computation for shorter completion times. We will add this discussion and plan to explore it in future work.
> > >
> > > Again, we sincerely appreciate your constructive insights. They have helped us refine both the analysis of execution gaps and the presentation of experimental details.

---

> > > > ### Comment · Reviewer_fjcC · 2025-08-07
> > > >
> > > > I thank the authors for the clarifications and have no further questions. I hope the relevant discussions on the planning/execution gap as well as accuracy of the system dynamics predictions (by `TimePredict`) in planning find their way to the revised version of the paper. As a final note, I would suggest the authors make the effort to open-source their implementation in a way that future work could include it as a baseline. Ideally, the STL specifications would be implemented in a standardized way and match other existing STL planning tools (e.g. [STLCG](https://github.com/UW-CTRL/stljax), [diff-spec](https://github.com/ZikangXiong/diff-spec)) to make comparisons easier but this is not a hard requirement.

---

> > > > > ### Author Response · Authors · 2025-08-08
> > > > >
> > > > > Thank you sincerely for your continued engagement and for the many constructive suggestions that have helped us improve the work. We will incorporate the expanded discussion on the planning–execution gap and on *TimePredict* accuracy into the revised manuscript.
> > > > >
> > > > > We plan to release our full implementation and experimental scripts on GitHub. Due to policy constraints, we cannot include any links here; however, the link will be provided in the revised manuscript. While doing so, we will make every effort to align our STL-specification interface with popular tools (e.g., STLCG, diff-spec) so that future research can compare against our baseline with minimal effort.
> > > > >
> > > > > Thank you once again for your thoughtful feedback and support.

---

### Official Review · Reviewer_TNZu · 2025-07-02

**Clarity:** 2
**Significance:** 3
**Originality:** 3
**Rating:** 5
**Confidence:** 4

**Summary:**

The authors propose a method for satisfying STL specifications using a combination of spatio-temporal decomposition and trajectory generation using diffusion. Using previously collected trajectory data, the authors generate estimates for travel times between regions in space, in addition to training a diffusion model for generating short trajectories that can be stitched together. The decomposed STL specification is used to generate “progresses” that capture reachability and invariance properties implied by the STL specification. The time estimates assign times to these progresses to generate a sequence of timed waypoints, and the diffusion model is used to generate trajectories to reach those timed waypoints. Results are compared against a recent diffusion approach for STL satisfaction.

**Questions:**

Was there a reason that alternatives to diffusion were not used as comparators? For example, one could imagine using RRT* or related methods to generate candidate trajectory segments. Since there needs to be some controller to track the diffusion trajectory anyway, it seems like you could apply the same controller to an RRT* generated trajectory.

Relatedly, what were the dynamics used in the simulation results? This seems to have strong implications for how successful the method would be. I would expect it to be easier to learn a model for a single integrator than for more complex nonlinear dynamics. However, the latter is where a method such as this would probably be more useful.

Where do $a_{\Lambda,\mathbb{T}}^{min}$ and $b_{\Lambda,\mathbb{T}}^{max}$ come from? I was confused about the heuristic order, and it seems to hinge on how these are calculated.

**Ethical Concerns:**

["NO or VERY MINOR ethics concerns only"]

**Final Justification:**

I am satisfied by the responses by the authors to my own questions, as well as to the questions posed by the other reviewers. Most importantly, I am glad that the authors have elected to include the robustness value. Therefore, I choose to keep my original rating of Accept.

**Limitations:**

Yes

**Quality:**

3

**Strengths And Weaknesses:**

Strengths:
- This paper covers an interesting and timely problem area. The method outperforms recent methods on the provided metrics.
- The discussion of limitations is thorough and fair, and the authors even include examples of failure cases.
- The paper is reasonably clear—for someone well-versed in this area, it is easy to gather the main ideas and most of the important details.

Weaknesses:
- The descriptions in sec. 3.3 should refer to line numbers in the algorithms. It seems like the backtracking element is not presented in the algorithm. Does this happen outside of Algorithm 1, or within it? In general, some more details about the algorithm would be appreciated – is it guaranteed to terminate? Obviously, the results suggest it is fast, but what is the complexity? If I’m not mistaken, there is an ILP in the algorithm that could cause bottlenecks depending on what variables impact its performance.
- Unless I missed it, there is no comparison of robustness of RGD and the proposed method. Quantitative semantics are one of the key motivators for using STL, and it would be informative to see whether the methods perform comparably.

Minor:
- Line 110, missing space in “unknown,but”
- Line 202, should “way-point” be “waypoints”?
- Line 215 “remained” should be “remaining”
- Line 252 “fucntion"
- Line 286 “estimates” should be “estimate”

---

> ### Author Rebuttal · Authors · 2025-07-31
>
> We scincerely thank you for the very constructive comments, especially for suggesting the inclusion of robustness as an additional quantitative metric and for pointing out the simplicity of our initial experimental settings. These suggestions prompted us to expand and strengthen our evaluation. We also appreciate your recognition of the novelty of our approach.
>
> **R3.1 Lack of detail in Algorithm 1: backtracking element not clear, termination guarantees**
>
> Back‑tracking is handled **within** Algorithm 1 via the explicit **stack**: Each loop (lines 5–18) pops the current node, iterates over its remaining reachability progresses, tries to sample a waypoint, updates constraints, then pushes the resulting child node. When a branch fails or is exhausted, the next `pop` naturally returns to a shallower depth and no external mechanism is required.
>
> For readability, the pseudo-code presents the search with an explicit stack, whereas our reference implementation executes the same logic recursively. We will clarify this in the revised manuscript.
>
> **Termination** is ensured because the search space is finite and monotonically shrinking: each successful expansion removes exactly one unsatisfied progress from $\mathbb{P}^{\mathcal{R}}$, and a sampling failure never re-pushes the same node. Hence the maximum depth equals the initial number of reachability progresses, and with finite branching the DFS must finish after exploring all configurations or upon finding a complete solution.
>
>
> **R3.2 Complexity, impact of ILP**
>
> A precise worst‑case analysis of the Progress Allocation search is indeed intractable because the branching factor varies with STL structure and time‑window interactions. In broad terms, runtime grows exponentially with the number of reachability progresses, yet *in practice* two factors keep it modest: (i) heuristic ordering and early pruning limit branching, and (ii) inter‑progress time‑window tightening rapidly collapses the feasible subtree. **Empirically the allocator runs in milliseconds–seconds** and is not the overall bottleneck. We will explore stronger heuristics and branch-parallelisation to further improve scalability in our future.
>
> About ILP:
> The **ILP** that arise inside Progress Allocation are lightweight and well-structured: (i) Each instance involves a *fixed* set of time variables (their count does not grow during search) with tight upper‑ and lower‑bound box constraints inherited from the STL time windows. (ii) All objectives and inequalities are 0‑1 combinations of those variables (iii) In scenarios involving repeated ILP solves, the objectives remains unchanged, and constraints are incrementally added rather than rebuilt from scratch. Consequently, each ILP solve is, on average, extremely fast.
>
> In our current implementation, we do not yet exploit these structural properties and instead rely on the off‐the‐shelf solver PuLP (a Python‐based LP/MIP modeler). Our experiments confirm that, **even with this straightforward setup, the ILP component does not constitute a performance bottleneck**. We will investigate in future work how these properties can be leveraged to further improve efficiency.
>
> **R3.3 No comparison of robustness**
>
> Following your suggestion, we computed the average robustness of the executed trajectories (excluding the top & bottom 5 %) for both methods on Maze2d. Our approach outperforms RGD in every scenario:
>
> |Env|Template|RGD|Ours|
> |-|-|-|-|
> |U | 1 | 0.1084 ± 0.1132  | 0.1938 ± 0.0715 |
> |U | 2 | -0.2208 ± 0.2826 | 0.1504 ± 0.0814 |
> |U | 3 | -0.1695 ± 0.2521 | 0.1354 ± 0.0745 |
> |M | 1 | 0.0885 ± 0.2768  | 0.3205 ± 0.0957 |
> |M | 2 | -0.2277 ± 0.3089 | 0.2013 ± 0.1950 |
> |M | 3 | -0.2393 ± 0.3130 | 0.1761 ± 0.2060 |
> |L | 1 | -0.1927 ± 0.3198 | 0.3180 ± 0.1228 |
> |L | 2 | -0.3926 ± 0.2284 | 0.1672 ± 0.2411 |
> |L | 3 | -0.2886 ± 0.2934 | 0.1639 ± 0.2286 |
>
> These results (will be added to the revised paper) reflect that the trajectories produced by our algorithm consistently better satisfy the STL tasks.
>
> **R3.4  Why alternatives to diffusion (RRT\*) were not used as comparators.**
>
> Classical planners such as RRT* require explicit knowledge of both the **environment map** and the **system dynamics**—neither is available in our setting. In Maze2d, the agent’s dynamics and the maze geometry are both *unknown*; this renders RRT* (and some optimization-based methods) inapplicable. By contrast, a diffusion model can be trained solely from offline trajectories, implicitly capturing dynamics and environmental constraints without needing a map. This is also one factor why RRT* is often not selected as a comparison baseline in many diffusion-based trajectory planning studies [1,2,3].
>
> **R3.5 What are the dynamics used in simulation? Why not more complex systems?**
>
> In the Maze2D experiments (Sec. 5), we control a spherical agent whose intrinsic dynamics are relatively simple, but the maze environment introduces complex constraints that also must be learned from trajectory data, thereby increasing the overall complexity. In the comparative experiments in Sec. G, we use double‐integrator dynamics.
>
> In response to your suggestion, we have **extended our evaluation to two dynamics‑rich tasks** from the offline goal‑conditioned RL benchmark **OGBench** [4]: *cube-single-play* (“cube”) and *antmaze-medium-navigate* (“antmaze”). The cube environment involves controlling a 6-DoF UR5e robot arm to manipulate cubes, whereas the antmaze environment involves controlling an 8-DoF Ant agent to navigate in a maze.
>
> In our experiments, we train exclusively on the STL task-agnostic trajectory datasets provided by OGBench, but replace the original goal-conditioned tasks with STL tasks. Specifically, in the antmaze environment, we adopt the same task formulation as in Sec. 5, requiring the agent to reach specified regions to satisfy the STL formula. In the cube environment, we focus solely on the end-effector position, requiring it to visit designated regions to satisfy the STL task (ignoring cube manipulation). In the cube environment, following the settings in our experiment, we first generate trajectories in the $x$–$y$–$z$ space and then track them with a PD controller. In the antmaze environment, following [1], we plan trajectories in the $x$–$y$ space and track with an inverse dynamics model that takes a 29-dimensional observation and the next $x$–$y$ goal as input and outputs an 8-dimensional action at each step.
>
> As in Sec. 5, we generate 100 test cases for each of the nine task templates (Table F.1) in both environments and report the execution success rates in the following Table. These results (over 60 % in antmaze and over 85 % in cube) confirm that our framework remains effective under substantially more complex dynamics.
>
> A more detailed description of the experimental setup, along with comprehensive results and analysis, will be provided in the revised manuscript.
>
> |Environment|Template|Execution Success Rate (%)|
> |-|-|-|
> |antmaze|1|**91.0**|
> |antmaze|2|**97.0**|
> |antmaze|3|**84.0**|
> |antmaze|4|**63.0**|
> |antmaze|5|**71.0**|
> |antmaze|6|**63.0**|
> |antmaze|7|**81.0**|
> |antmaze|8|**87.0**|
> |antmaze|9|**67.0**|
> |cube|1|**98.0**|
> |cube|2|**100.0**|
> |cube|3|**98.0**|
> |cube|4|**98.5**|
> |cube|5|**85.0**|
> |cube|6|**90.0**|
> |cube|7|**97.0**|
> |cube|8|**100.0**|
> |cube|9|**99.0**|
>
>
> Our current implementation adopts **simple baseline modules** for time prediction, trajectory generation, and low-level control. In highly challenging scenarios such as **antmaze**, these basic choices become potential performance bottlenecks. Since our framework is designed to be modular, each component can be replaced by a stronger alternative. Recent advances in diffusion-based trajectory planning (e.g., [1, 2]) provide promising drop-in candidates. We will integrate these enhanced modules and extend the evaluation to even more demanding environments to quantify their benefit in future work.
>
> **R3.6 Clarify $a_{\Lambda,\mathbb{T}}^{min}$ and how heuristic order is computed.**
>
> For a given progress $\mathcal{P}(a_\Lambda, b_\Lambda, \mu)$, $a_{\Lambda,\mathbb{T}}^{min}$ denotes the minimal possible value of $a_\Lambda$ under the current time‐variable constraints in $\mathbb{T}$. It reflects the earliest possible activation time of that progress. It can be obtained by solving the following ILP:
> \begin{aligned}
> &\min_{\lambda} \quad a_\Lambda(\lambda) \\
> \text{s.t.}\quad & \lambda \models \mathfrak{t}, \quad \forall \mathfrak{t} \in \mathbb{T},\\
> & \lambda \in \mathbb{Z}_{+}^{|\Lambda|}.
> \end{aligned}
>
> The quantities $a_{\Lambda,\mathbb{T}}^{max}$, $b_{\Lambda,\mathbb{T}}^{min}$, and $b_{\Lambda,\mathbb{T}}^{max}$ are defined analogously. Due to space limitations, we have provided only a brief description of the above at lines 230–235; the revised version will include a more detailed exposition.
>
> Regarding the Heuristic Order (lines 283–285), we prioritize progresses with earlier potential deadlines $b_{\Lambda,\mathbb{T}}^{min}$. If multiple candidates share the same deadline, we prefer the one with earlier potential start time $a_{\Lambda,\mathbb{T}}^{min}$.
>
> **R3.7 Typos and descriptions in Sec. 3.3**
>
> Thank you for spotting these issues. We will fix all reported typos and revise Sec. 3.3 so that the exposition is clearer in the final version.
>
> > [1] Luo Y, Mishra U A, Du Y, et al. Generative trajectory stitching through diffusion composition[J]. arXiv:2503.05153, 2025.
> >
> > [2] Chen C, Hamed H, Baek D, et al. Extendable long-horizon planning via hierarchical multiscale diffusion[J]. arXiv:2503.20102, 2025.
> >
> > [3] Xiao W, Wang T H, Gan C, et al. Safediffuser: Safe planning with diffusion probabilistic models[C]//The Thirteenth International Conference on Learning Representations.
> >
> > [4] Park S, Frans K, Eysenbach B and Levine S (2025) Ogbench: Benchmarking offline goal-conditioned rl. In: International Conference on Learning Representations.

---

### Official Review · Reviewer_E3cN · 2025-07-03

**Clarity:** 2
**Significance:** 2
**Originality:** 2
**Rating:** 4
**Confidence:** 3

**Summary:**

The paper seeks to generate plans that satisfy signal temporal logic (STL) specifications without access to a transition model but assumes the state variables and their domain (values that can be assigned to them) are known. The algorithm decomposes an STL specification into a set of reachability and invariance progresses as well as time constraints. Based on this and a start state, a depth-first search is performed for a sequence of waypoints that satisfies the STL. At each node in the depth-first search, all possible unsatisfied progresses are examined and states that satisfy those progresses are sampled. After the waypoints are found a diffusion model trained on offline data is used to find the intermediate states and a sequence of actions that connects the state sequence is found via an inverse dynamics model. The algorithm was evaluated on 2D maze tasks.

**Questions:**

How is sampling of x’ in Algorithm 2 performed such that x’ entails \mu?

**Ethical Concerns:**

["NO or VERY MINOR ethics concerns only"]

**Final Justification:**

The authors addressed my concerns about search scaling. The details on the inverse model, which I am now told is trained, are not clear. So, I am adjusting my score from a 3 to a 4.

**Limitations:**

Search for waypoints is based on a depth-first search, which will not scale to larger state spaces.
Sampling waypoints does not directly depend on the current waypoint, which can lead to inefficient plans.

**Quality:**

2

**Strengths And Weaknesses:**

**Strengths**

The approach can find a sequence of states that satisfies an STL specification without needing a model since the diffusion model is trained via an offline dataset. Decomposes and STL specification into progresses to better integrate with search for waypoints.

**Weaknesses**

It appears the main innovation of this paper is satisfying STL constrains in a model-free manner. However, the algorithm to accomplish this finds waypoints via a depth-first search that does not use a domain-specific heuristic function (which could be learned). Therefore, I do not believe this algorithm will scale to larger search spaces.

While there is preference given to next waypoints with a lower predicted time to be reached from the current waypoint, the manner in which these states are generated does not directly depend on the current waypoint. Therefore, very inefficient plans may be found.

Given the sequence of states, generating action sequences still depends on an inverse model. Therefore, I do not think that this method truly addresses the problem of satisfying STL plans with unknown dynamics.

For the evaluation, there are failure cases. However, it is not clear where the algorithm failed (i.e. at waypoint search, state trajectory generation with the diffusion model, or action generation).

---

> ### Author Rebuttal · Authors · 2025-07-31
>
> We sincerely thank you for the very constructive comments, particularly for highlighting the gaps in our analysis of failure modes. This insight has guided us to provide a more detailed breakdown of where and why our method may fail, making our analysis more comprehensive.
>
> **R2.1 The method lacks scalability due to naive DFS and no learned domain-specific heuristic.**
>
> Our DFS-style search is not a blind traversal. In fact, we employ two kinds of domain-specific heuristics: task-aware and environment-aware.
> - For the task-aware heuristic, at each iteration we order the remaining progresses using the **Heuristic Order** (lines 283–285), prioritizing those with tighter admissible time windows.
> - The environment-aware heuristic is implicitly embedded via the TimePredictor, which is trained on abundant trajectory data consistent to the actual behavior in the environment, thereby assisting in determining the timing of each waypoint and guiding the search.
>
> In addition to these heuristics,
> **(i)** We apply an **early-pruning rule** (lines 230–239) that backtracks immediately when the feasible set of shared time variables becomes empty.
> **(ii)** The search is conducted **in the space of reachability progresses**, not in the full product space of task and workspace. Workspace states are derived directly and efficiently from the corresponding states in the progress space.
>
> Empirically, the Progress Allocation module accounts for only a small fraction of the total runtime (Sec. 5). Even for complex, long-horizon STL tasks with multi-layer nested structures that the optimization-based counterpart struggles to solve efficiently (Sec. G), this module does not become a performance bottleneck. Furthermore, in our additional tests on two scenarios with more complex dynamics with a 6-DoF robotic arm and an 8-DoF ant agent (see Sec. R1.3 in response to reviewer vwWP), the module still find a solution with an average time of less than 1s and does not pose a performance bottleneck. These experiments demonstrate that, with the aforementioned optimizations, our Progress Allocation module exhibits strong scalability.
>
> We agree that other learned heuristics may further improve scalability, and we will explore this direction in future work.
>
>
> **R2.2 Candidate samples of next waypoint are drawn without conditioning on the current waypoint, inefficient plans may be found.**
>
> The main objective of this work is to ensure dynamic feasibility rather than optimality. Therefore, our current method for obtaining the next waypoint involves sampling a candidate state within the target region and checking whether it is dynamically feasible to reach that state within the admissible time window from the current waypoint. As a result, compared to selecting the “nearest” state, the resulting trajectory may indeed be longer. Of course, we could incorporate additional heuristics to preferentially sample states that are as “close” as possible. However, we believe that pursuing a globally optimal (shortest) trajectory within our current framework is a highly challenging problem: relying solely on this myopic strategy is insufficient and would likely require further global structural adjustments. In the revised manuscript, we will clarify that our focus is on dynamic feasibility rather than optimality, and we plan to explore approaches for optimality in future work.
>
>
> **R2.3 Action generation still depends on an inverse dynamics model, doesn’t fully solve unknown‑dynamics planning.**
>
> The inverse-dynamics model is not given but  trained **offline** on the trajectory data that implicitly encode system dynamics. Therefore, it requires **no analytic model**. This aligns with the unknown-dynamics setting we consider here. In fact, using an inverse-dynamics model as the low-level controller to generate action sequences is a common practice in diffusion-based trajectory planning frameworks, and our approach follows this standard methodology [1,2].
>
>
> **R2.4 Failure‑case analysis is missing, it’s unclear which stage fails.**
>
> Our experiments reveal three distinct failure modes:
> 1. **Allocation failure.** The Progress Allocation algorithm can occasionally miss an existing solution because it is not complete. This is quantified by the **Progress Allocation Success Rate** in Sec. G and discussed with remedies in Sec. B (*Limitations*).
> 2. **Execution failure.** The planned trajectory satisfies the STL but is difficult to track under real dynamics by the controller, leading to task violation during execution. This is the **dominant** error source and is captured by the **Execution Success Rate** in Secs. 5 & G; mitigation ideas (e.g., online replanning) appear in Sec. F.1.
> 3. **Rare synthesis failure.** Our method provides a formal guarantee for generating STL‑compliant trajectories (Theorem 1), with part of the guarantee inherited from SafeDiffuser [3]. In very rare cases, SafeDiffuser fails to generate a segment that fully respects the constraints, yielding an STL‑violating plan; across **4050** Maze2d trials, this occurred only **5** times (≈ **0.12 %**).
>
> We will add a dedicated subsection summarizing these statistics, clarifying where failures occur, and outlining mitigations for each mode.
>
> **R2.5 Algorithm 2: how is `x'` sampled such that it satisfies μ?**
>
> As detailed in Sec. 2.2, each atomic predicate $\mu$ comes with an *explicit* evaluation function $h_{\mu}$. A state $x$ satisfies $\mu$ iff $h_{\mu}(x) \geq 0$. Consequently, one can perform rejection sampling (or uniform sampling over a pre-computed feasible set) and accept a candidate $x'$ only when $h_{\mu}(x') \geq 0$. Because $h_{\mu}$ is part of the task specification, this check is analytic and incurs negligible overhead.
>
> > [1] Ajay A, Du Y, Gupta A, et al. Is Conditional Generative Modeling all you need for Decision Making?[C]//The Eleventh International Conference on Learning Representations.
> >
> > [2] Luo Y, Mishra U A, Du Y, et al. Generative trajectory stitching through diffusion composition[J]. arXiv:2503.05153, 2025.
> >
> > [3] Xiao W, Wang T H, Gan C, et al. Safediffuser: Safe planning with diffusion probabilistic models[C]//The Thirteenth International Conference on Learning Representations.

---

> > ### Author Response · Authors · 2025-08-08
> >
> > Dear Reviewer E3cN,
> >
> > Thank you once again for your time and thoughtful review of our paper. We sincerely appreciate the constructive feedback you provided, which has helped us improve our work.
> >
> > As mentioned in our previous message, we have carefully addressed your concerns by clarifying the requested points and conducting additional, more complex experiments. **These new experimental results demonstrate that our method scales well even in more challenging settings**.
> >
> > Since today marks the end of the rebuttal period and we have not yet received your updated feedback, we wanted to kindly check if our clarifications meet your expectations or if any further explanation is needed. We would be more than happy to provide additional details or address any remaining concerns you might have.
> >
> > Thank you again for your time and valuable input. Please let us know if there is anything further we can clarify.
> >
> > Best Regards,
> >
> > Authors of Paper 11376

---

### Official Review · Reviewer_vwWP · 2025-07-03

**Clarity:** 3
**Significance:** 1
**Originality:** 3
**Rating:** 3
**Confidence:** 3

**Summary:**

This paper presents a planning algorithm for STL specifications. The proposed method first decomposes the STL formula into several basic formulations with reachability, invariance, and temporal constraints. Then a search-based algorithm is developed to generate waypoints that can satisfy this STL. The full trajectory is generated with a conditional diffusion model pre-trained on trajectory data.

**Questions:**

Please see the weaknesses part.

**Ethical Concerns:**

["NO or VERY MINOR ethics concerns only"]

**Final Justification:**

After the rebuttal and discussion, I acknowledge the clarifications provided regarding the additional experiments. The main contribution of the machine learning module is a time predictor, which, however, faces limitations on sparse datasets and is prone to overfitting. Including evaluations in a high-dimensional task space would further strengthen the contribution.

**Limitations:**

Yes

**Quality:**

2

**Strengths And Weaknesses:**

## Strengths

* The problem of planning for STL from offline trajectory data is interesting and meaningful.

* The proposed decomposition and search algorithm can solve certain STL formulations.

## Weaknesses

* Major

  * The major weakness is that the proposed search algorithm for waypoints generation did not take system dynamics into consideration. It seems that from STL decomposition to allocation, the proposed solution depends mainly on the STL formulation itself. Although there is a neural network based time predictor that can predict the time required to visit a future state from a current state, this predictor can overfit or generate unrealistic predictions. My understanding is that if two states are not in the same trajectory in the dataset, then according to the description of this training pipeline, the time predictor cannot learn state pairs outside individual episodes.
  * Therefore:
    * In state search/sample, it might not find new state that is actually possible by stitching together dataset trajectories;
    * The generated waypoints, although can satisfy the STL, may not be achievable due to overfitting, as there does not exist a valid trajectory to satisfy the goal based on demonstration;
    * Although scaling factors are added for time prediction, this does not align well with the actual trajectories, as the avoidance requirements may lead to much longer distance and time through detour path.
  * The experimental environments used in this work are very toy example as they are all two dimensional navigation tasks for a point mass agent.

* Minor

  * The proposed framework depends on additional assumptions on the form of STL, for example, assuming operator “until” only has “always” temporal operator for its first operand, assuming DNF only has single subformula, and using replacement in DNF. The method only solves the resulting STL. Therefore, the solution is no longer optimal with respect to the original STL specification. And in the worst case, there might be no feasible solutions after these assumptions and transformation.

  * The paper also implicitly assumes that the evaluation function is know to the planner if I am understanding correctly. I think this should be pointed out since it will make the method unrealistic in the environments where the atomic propositions of a state can only be observed once the agent really interacts that state.

---

> ### Author Rebuttal · Authors · 2025-07-31
>
> We sincerely thank you for the constructive feedback, especially for pointing out the limitations of our experimental scenarios. This insight motivated us to expand our experiments and strengthen our empirical evaluation.
>
> **R1.1 Major: The search algorithm for waypoints does not consider system dynamics.**
>
> Our waypoint search **does** incorporates dynamic feasibility **implicitly** through the **TimePredictor**, which is trained on a large set of offline trajectories generated under the actual system dynamics. By estimating realistic travel times between states, the TimePredictor determines the timing of each waypoint, ensuring feasible transitions and filtering out dynamically infeasible ones.
>
> The TimePredictor’s dynamic information is leveraged during the search process as follows: without dynamics constraints, the search algorithm would produce a vast number of allocations due to unconstrained variable choices. However, by integrating the TimePredictor into the search, we steer solutions toward dynamically feasible ones, and effectively use the implicit system dynamics to guide the search.
>
> For the unknown dynamic problem setting, in principle,  one can never guarantee **absolute** dynamic compliance as only latent information can be inferred from trajectory data. Nevertheless, our experiments show that the algorithm still successfully finds dynamically feasible solutions in most cases. Even in the more complex antmaze scenarios (detailed in R1.3), where our method maintains strong feasibility performance.
>
> **R1.2 Major: The time predictor may over-fit or fail for unconnected states; the method may miss possible samples or produce unreachable plans, especially when avoidance enlarges path length.**
>
> The role of the TimePredictor is to leverage historical trajectory data to estimate the realistic travel time between two states. In scenarios with abundant historical trajectories, most state pairs are covered by these data, so the TimePredictor can make accurate predictions in most cases, providing a degree of feasibility assurance for the final solutions, as reflected in our experimental results.
>
> We acknowledge that when data are sparse, some state transitions may not be represented in the trajectories, causing the TimePredictor to produce unrealistic predictions, particularly in cases that require avoidance behaviors. We will include a discussion of this limitation in the revised version.
>
> Currently, we mainly focus on the scenario with abundant historical trajectories and begin with a lightweight TimePredictor module and compensate by exposing a redundancy scaling factor and already yields a relative high success rates. In situations with limited trajectory data, since our framework is modular, we allow the TimePredictor module to be improved, e.g., by swapping in a more powerful model or by employing trajectory-stitching data-augmentation techniques (e.g., [1]) to generate longer cross-episode samples. We plan to explore these improvements in future work.
>
>
> **R1.3 Major: The experimental environments used in this work are simple.**
>
> We have **extended our evaluation to two dynamics‑rich tasks** from the offline goal‑conditioned RL benchmark **OGBench** [2]: *cube-single-play* (“cube”) and *antmaze-medium-navigate* (“antmaze”). The cube environment involves controlling a 6-DoF UR5e robot arm to manipulate cubes, whereas the antmaze environment involves controlling an 8-DoF Ant agent to navigate in a maze.
>
> In our experiments, we train exclusively on the STL task-agnostic trajectory datasets provided by OGBench, but replace the original goal-conditioned tasks with STL tasks. Specifically, in the antmaze environment, we adopt the same task formulation as in Sec. 5, requiring the agent to reach specified regions to satisfy the STL formula. In the cube environment, we focus solely on the end-effector position, requiring it to visit designated regions to satisfy the STL task (ignoring cube manipulation). In the cube environment, following the settings in our experiment, we first generate trajectories in the $x$–$y$–$z$ space and then track them with a PD controller. In the antmaze environment, following [3], we plan trajectories in the $x$–$y$ space and track with an inverse dynamics model that takes a 29-dimensional observation and the next $x$–$y$ goal as input and outputs an 8-dimensional action at each step.
>
> As in Sec. 5, we generate 100 test cases for each of the nine task templates (Table F.1) in both environments and report the execution success rates in the following Table. These results (over 60 % in antmaze and over 85 % in cube) confirm that our framework remains effective under substantially more complex dynamics.
>
> A more detailed description of the experimental setup, along with comprehensive results and analysis, will be provided in the revised manuscript.
>
> |Environment|Template|Execution Success Rate (%)|
> |-|-|-|
> | antmaze | 1 | **91.0**    |
> | antmaze | 2 | **97.0**    |
> | antmaze | 3 | **84.0**    |
> | antmaze | 4 | **63.0**    |
> | antmaze | 5 | **71.0**    |
> | antmaze | 6 | **63.0**    |
> | antmaze | 7 | **81.0**    |
> | antmaze | 8 | **87.0**    |
> | antmaze | 9 | **67.0**    |
> | cube| 1 | **98.0**    |
> | cube| 2 | **100.0**   |
> | cube| 3 | **98.0**    |
> | cube| 4 | **98.5**    |
> | cube| 5 | **85.0**    |
> | cube| 6 | **90.0**    |
> | cube| 7 | **97.0**    |
> | cube| 8 | **100.0**   |
> | cube| 9 | **99.0**    |
>
> Our current implementation adopts **simple baseline modules** for time prediction, trajectory generation, and low-level control. In highly challenging scenarios such as **antmaze**, these basic choices become potential performance bottlenecks. Since our framework is designed to be modular, each component can be replaced by a stronger alternative. Recent advances in diffusion-based trajectory planning (e.g., [1, 3]) provide promising drop-in candidates. We will integrate these enhanced modules and extend the evaluation to even more demanding environments to quantify their benefit in future work.
>
> **R1.4 Minor: Assumptions on STL (e.g., restricted until, DNF transformation)**
>
> The extra syntactic restrictions and transformations are **intentional**: they enable tractable progress decomposition and simplify allocation. While they narrow the theoretical task class, they still cover *the vast majority of practical robotic tasks specifications*. In our work, the formula in Disjunctive Normal Form (DNF) is slightly stronger than the original expression. However, converting the original expression into this form improves scalability while preserving soundness: it may reject some borderline-feasible tasks, but it never produces a false-positive plan. This trade-off is discussed in Sec. B (*Limitations*).
>
> Similar simplifications and assumptions are common in STL planning or control synthesis literature (e.g., [4, Sec. III-A] and [5, Remark 2.2–2.3]) and have proven effective in real-world deployments.
>
> **R1.5 Minor: Implicitly assumes the evaluation function (for APs) is known during planning.**
>
> We focus on scenarios in which the evaluation functions for atomic propositions (APs) are provided alongside the STL task. In some settings an agent must explore a specific state before it can determine whether a particular AP is satisfied. However, those cases primarily involves **planning and exploration** in unknown environments, which differ from our scenario assumptions. In the revised manuscript, we will further discuss the distinctions between our scenario and those settings.
>
> > [1] Chen C, Hamed H, Baek D, et al. Extendable long-horizon planning via hierarchical multiscale diffusion[J]. arXiv:2503.20102, 2025.
> >
> > [2] Park S, Frans K, Eysenbach B and Levine S (2025) Ogbench: Benchmarking offline goal-conditioned rl. In: International Conference on Learning Representations.
> >
> > [3] Luo Y, Mishra U A, Du Y, et al. Generative trajectory stitching through diffusion composition[J]. arXiv:2503.05153, 2025.
> >
> > [4] Yu P, Tan X, Dimarogonas D V. Continuous-time control synthesis under nested signal temporal logic specifications[J]. IEEE Transactions on Robotics, 2024, 40: 2272-2286.
> >
> > [5] Yu P, Gao Y, Jiang F J, et al. Online control synthesis for uncertain systems under signal temporal logic specifications[J]. The international journal of robotics research, 2024, 43(6): 765-790.

---

> > ### Author Response · Authors · 2025-08-06
> >
> > Dear Reviewer vwWP,
> >
> > Thank you very much for your thoughtful feedback and the time you dedicated to evaluating our submission.
> >  We have carefully addressed each of your suggestions, particularly, including additional experiments in more challenging scenarios.
> >
> > As the discussion period is nearing its end, we would greatly appreciate any indication of whether our revisions satisfactorily clarified the concerns you raised or if there are further aspects we could refine. Any additional guidance you could provide would be very valuable to us.
> >
> > Thank you again for your time and consideration.
> >
> > Sincerely,
> > The Authors of Paper 11376

---

> > ### Author Response · Authors · 2025-08-08
> >
> > Dear Reviewer vwWP,
> >
> > Thank you once again for your time and thoughtful review of our paper. We sincerely appreciate the constructive feedback you provided, which has helped us improve our work.
> >
> > As mentioned in our previous message, we have carefully addressed your concerns by clarifying the requested points and conducting additional, more complex experiments. These new results demonstrate that our method performs robustly even in more challenging settings.
> >
> > Since today marks the end of the rebuttal period and we have not yet received your updated feedback, we wanted to kindly check if our clarifications meet your expectations or if any further explanation is needed. We would be more than happy to provide additional details or address any remaining concerns you might have.
> >
> > Thank you again for your time and valuable input. Please let us know if there is anything further we can clarify.
> >
> > Best Regards,
> >
> > Authors of  Paper 11376

---

> > > ### Comment · Reviewer_vwWP · 2025-08-08
> > >
> > > I would like to thank the authors for their detailed responses and explanations. My concern regarding the use of a TimePredictor remains as one limitation of TimePredictor is that it stills rely on abundant offline trajectories to cover all possible paired states. I appreciate the authors for providing extra experiments in new environments. However, these environments are also 2d/3d navigation-like without manipulation since the control mostly rely on an inverse model. The complexity and reliability of learning a TimePredictor might increase in high-dimensional planning space, for example when the environment has events that depend on additional information of the robot other than end-effector coordinates.

---

> > > > ### Author Response · Authors · 2025-08-09
> > > >
> > > > Thank you for engaging further with our work and for the constructive suggestions; your comments have helped us refine both the scope and the presentation.
> > > >
> > > > **Scope and setting.**
> > > > Our study targets *temporal-logic task planning* where specifications are typically posed in a low-dimensional **task space** (2D/3D regions and relations). Planning is therefore conducted in task space, and a low-level controller executes the plan on the full system. We adopt this decoupling for two reasons:
> > > >
> > > > 1) in many practical applications the task itself is naturally defined in workspace coordinates rather than configuration space;
> > > > 2) under unknown dynamics, ensuring STL satisfaction is already highly challenging even in task space.
> > > >
> > > > Empirically, our results show strong execution performance even when the underlying dynamics are complex (e.g., ant-like locomotion and 6-DoF arm tracking). This setting follows a common evaluation protocol in the temporal-logic planning/control-synthesis literature, which often uses navigation-style tasks regardless of whether dynamics are known [1~7]. We also acknowledge recent work that plans **directly in high-dimensional configuration space** (with known dynamics) [8]; extending our framework to that regime is an important direction we plan to investigate.
> > > >
> > > > **On TimePredictor coverage.**
> > > > We agree that coverage becomes more demanding as the planning space’s dimensionality grows. Our current scope assumes **sufficient offline trajectory data**, a requirement shared by both the TimePredictor and the diffusion-based trajectory generator. Within this data regime, the TimePredictor has not been a bottleneck in our experiments. For settings with limited data or when planning directly in configuration space, our **modular** pipeline allows stronger estimators to be swapped in, and we can employ **trajectory-stitching data augmentation** to create cross-episode samples that improve coverage and generalization (see R1.2 and [9]). We will expand these clarifications in the revised manuscript.
> > > >
> > > > We appreciate your thoughtful feedback and will incorporate the above points in the revision.
> > > >
> > > >
> > > >
> > > > > [1] Feng Z, Luan H, Goyal P, et al. LTLDoG: Satisfying temporally-extended symbolic constraints for safe diffusion-based planning[J]. IEEE Robotics and Automation Letters, 2024.
> > > > >
> > > > > [2] Feng Z, Luan H, Ma K Y, et al. Diffusion meets options: Hierarchical generative skill composition for temporally-extended tasks[J]. arXiv:2410.02389, 2024.
> > > > >
> > > > > [3] Meng Y, Fan C. TeLoGraF: Temporal Logic Planning via Graph-encoded Flow Matching[J]. arXiv:2505.00562, 2025.
> > > > >
> > > > > [4] Tian D, Fang H, Yang Q, et al. Two-phase motion planning under signal temporal logic specifications in partially unknown environments[J]. IEEE Transactions on Industrial Electronics, 2022, 70(7): 7113-7121.
> > > > >
> > > > > [5] Yu P, Tan X, Dimarogonas D V. Continuous-time control synthesis under nested signal temporal logic specifications[J]. IEEE Transactions on Robotics, 2024, 40: 2272-2286.
> > > > >
> > > > > [6] Yu P, Gao Y, Jiang F J, et al. Online control synthesis for uncertain systems under signal temporal logic specifications[J]. The international journal of robotics research, 2024, 43(6): 765-790.
> > > > >
> > > > > [7] Kapoor P, Kang E, Meira-Góes R. Safe planning through incremental decomposition of signal temporal logic specifications[C]//NASA Formal Methods Symposium. Cham: Springer Nature Switzerland, 2024: 377-396.
> > > > >
> > > > > [8] Kurtz V, Lin H. Temporal logic motion planning with convex optimization via graphs of convex sets[J]. IEEE Transactions on Robotics, 2023, 39(5): 3791-3804.
> > > > >
> > > > > [9] Luo Y, Mishra U A, Du Y, et al. Generative trajectory stitching through diffusion composition[J]. arXiv:2503.05153, 2025.

---

### Note · Authors · 2025-08-12

Thank you for your efforts during the review and rebuttal, and for the helpful suggestions that improved our work. In the revision, we clarified our contributions and added more complex experiments. While some reviewers acknowledged the improvements, remaining comments mainly concern our problem setting; we therefore further clarify it and its significance.

**Problem setting and contribution.**
We study zero-shot STL planning under unknown system dynamics. As noted by the reviewer, “*this problem is interesting and meaningful*". The setting is challenging, as it must balance a newly given STL task over long horizon while ensuring feasibility under unknown dynamics. Prior work either tackles only simple point to point navigation (e.g. SafeDiffuser) or underperforms on complex STL tasks (e.g., RGD). We propose a modular decomposition–allocation–generation framework that, as noted by the reviewer, “*avoids the problem of getting long range trajectories*", "*much faster than existing approaches*”
and “*elegantly handles the slow inference of diffusion models*”.

**Task space vs. configuration space.**
A key question is scalability to high dimensional planning spaces. As in most of existing temporal logic planning works, our planning is performed in the **task space**, not the **configuration space**. Thus, our newly added experiments are already very complex, and comparisons with existing works are fair. To our knowledge, direct STL planning in high dimensional configuration space under unknown dynamics has not been addressed and likely exceeds the current state of the art.

**Abundant Data needs for TimePredictor.**
The data requirement is standard in planning with unknown dynamics; even without STL tasks, diffusion-based planners require substantial data. Our benchmarks provide such data, and TimePredictor performs well. We will explore improvements such as stitching-based data augmentation in future work.

**Execution details.**
Tracking uses a PD controller or an inverse dynamics model trained on the same trajectory data as the diffusion planner, without any analytic dynamics model. This is common in diffusion-based planning and fits the unknown-dynamics assumption. Execution is strictly time synchronized (advance by time, not by waypoint proximity). Performance is primarily determined by the dynamic feasibility of the planned trajectory.

Thank you for your time and consideration. We hope our final remarks assist the final decision.

---

### Decision · Program_Chairs · 2025-09-17

**Decision:**

Accept (poster)

**Comment:**

The authors propose a method for satisfying Signal Temporal Logic (STL) specifications through a combination of spatio-temporal decomposition and trajectory generation using diffusion. The paper addresses an interesting and timely problem and demonstrates that the proposed method outperforms recent approaches on the reported metrics. The discussion of limitations is thorough and balanced, with illustrative examples of failure cases provided. The manuscript is clear and well-structured. Overall, this work represents a valuable and timely contribution to the field, and with its innovative approach and strong empirical results, it merits publication.